# Autoinhibition of Bruton's tyrosine kinase (Btk) and activation by soluble inositol hexakisphosphate

Qi Wang[1,2], Erik M Vogan[3,4], Laura M Nocka[5], Connor E Rosen[1,2], Julie A Zorn[1,2], Stephen C Harrison[4]*, John Kuriyan[1,2,5,6]*

[1]Department of Molecular and Cell Biology, Howard Hughes Medical Institute, University of California, Berkeley, Berkeley, United States; [2]California Institute for Quantitative Biosciences, University of California, Berkeley, Berkeley, United States; [3]Beryllium Inc, Boston, United States; [4]Laboratory of Molecular Medicine, Harvard Medical School, Howard Hughes Medical Institute, Boston, United States; [5]Department of Chemistry, University of California, Berkeley, Berkeley, United States; [6]Physical Biosciences Division, Lawrence Berkeley National Laboratory, Berkeley, United States

**Abstract** Bruton's tyrosine kinase (Btk), a Tec-family tyrosine kinase, is essential for B-cell function. We present crystallographic and biochemical analyses of Btk, which together reveal molecular details of its autoinhibition and activation. Autoinhibited Btk adopts a compact conformation like that of inactive c-Src and c-Abl. A lipid-binding PH-TH module, unique to Tec kinases, acts in conjunction with the SH2 and SH3 domains to stabilize the inactive conformation. In addition to the expected activation of Btk by membranes containing phosphatidylinositol triphosphate ($PIP_3$), we found that inositol hexakisphosphate ($IP_6$), a soluble signaling molecule found in both animal and plant cells, also activates Btk. This activation is a consequence of a transient PH-TH dimerization induced by $IP_6$, which promotes transphosphorylation of the kinase domains. Sequence comparisons with other Tec-family kinases suggest that activation by $IP_6$ is unique to Btk.

*For correspondence: harrison@crystal.harvard.edu (SCH); jkuriyan@mac.com (JK)

## Introduction

Activation of B- and T-lymphocytes relies on a chain of tyrosine phosphorylation events initiated by B-cell and T-cell receptors, respectively (*Cambier et al., 1993*; *Weiss and Littman, 1994*; *Kurosaki, 2011*). These phosphorylation-dependent signals are generated by three kinds of non-receptor tyrosine kinases, linked to the receptor by membrane co-localization (*Dal Porto et al., 2004*; *Kurosaki, 2011*). Src-family kinases (principally Lyn and Fyn in B-cells, Lck and Fyn in T-cells), resident at the plasma membrane, respond first to receptor activation. They phosphorylate pairs of tyrosine residues in ITAM motifs in the cytoplasmic tails of activated receptors (*Chu et al., 1998*; *Mócsai et al., 2010*; *Wang et al., 2010*), which in turn recruit the tandem-SH2 domain tyrosine kinases Syk (in B-cells) and ZAP-70 (in T-cells). Membrane localization of Syk or ZAP-70 leads to the recruitment and activation of Tec-family tyrosine kinases, Btk (in B-cells) and Itk (in T-cells) (*Andreotti et al., 2010*).

Btk is present in all hematopoietic cells, except for T-cells and natural killer cells (*Tsukada et al., 1993*; *Lindvall et al., 2005*). It is also part of several other receptor-mediated signaling networks, including those initiated by Toll-like receptors (*Jefferies et al., 2003*), Fc receptors (*Kawakami et al., 1994*), and G-protein coupled receptors (*Tsukada et al., 1994*; *Bence et al., 1997*). Btk is critical for the generation of calcium flux in response to receptor activation, because it activates phospholipase C-γ2 (PLC-γ2), which produces the second messengers diaceylglycerol (DAG) and inositol

**eLife digest** The human immune system is our defense against a variety of intruders, including bacteria, viruses, and other microbes, and provides our body with long-lasting protection against these invaders. B-cells—one type of immune cell—produce proteins called antibodies that can recognize microbes from previous invasions. This allows other immune cells to find the intruders and destroy them more efficiently.

People suffering from a genetic disease called X-linked agammaglobulinemia have fewer fully mature B-cells than most people and are extremely susceptible to bacterial infections. The disease is caused by mutations in the gene encoding an enzyme called Bruton's tyrosine kinase (Btk), which is involved in B-cell development. This enzyme is usually kept inactive until the B-cells are needed, but the molecular mechanism of this restraint remains to be discovered.

Wang et al. have used X-ray crystallography and biochemistry to produce a three-dimensional atomic model of the Btk enzyme in its inactive form. The model and functional studies reveal that when the Btk enzyme is inactive, it folds up into a compact shape. Within the enzyme, a 'PH-TH module' and two other domains work together to stabilize this folded-up form.

Although earlier studies indicated that the Btk enzyme becomes activated only when it is recruited to cell membranes, Wang et al. found that a small molecule called $IP_6$ (short for inositol hexakisphosphate) can activate the Btk enzyme even in the absence of a membrane. This molecule recognizes a previously unnoticed binding site on the PH-TH module, and causes the PH-TH module in the inactive Btk enzyme to bind briefly to the PH-TH modules in other Btk enzymes. This brief binding event is an important step in the process that activates the enzyme. $IP_6$ might also be involved in recruiting the Btk enzyme to the cell membrane in B cells, but this proposal needs to be tested experimentally. The details of the BTK activation mechanism reported by Wang et al. could prove useful to researchers developing new treatments for immunodeficiency diseases.

(1,4,5)-trisphosphate ($IP_3$). Btk dysfunction is the cause of X-linked agammaglobulinemia, a severe disease of primary immunodeficiency (*Tsukada et al., 1993*).

The Tec family kinases have a lipid-interaction module attached to the N terminus of a Src-like module (i.e., concatenation of an SH3 domain, an SH2 domain and a tyrosine kinase domain) (*Figure 1A*). Like the non-receptor tyrosine kinase c-Abl, the Tec kinases lack a C-terminal tyrosine phosphorylation site that in Src kinases engages the SH2 domain in an autoinhibitory interaction (*Sicheri et al., 1997*; *Xu et al., 1997*; *Nagar et al., 2003*). The lipid-interaction module of the Tec kinases is an extended plekstrin homology (PH) domain fused to a Tec homology (TH) domain (*Hyvönen and Saraste, 1997*). The PH-TH binds $PIP_3$ in membranes, or the soluble head group of $PIP_3$, inositol tetrakisphosphate ($IP_4$), in solution. A flexible linker connects the PH-TH module to the Src-like module; its length varies from 26 residues in Itk, the Tec family kinase in T-cells, to 45 residues in Btk.

Tyrosine phosphorylation controls Btk activity. B-cell activation leads to rapid phosphorylation of Tyr 551 in the kinase domain and Tyr 223 in the SH3 domain of Btk (*Park et al., 1996*; *Rawlings et al., 1996*). Tyr 551 corresponds to the canonical phosphorylation site in the activation loop of tyrosine kinases; as expected, its phosphorylation increases Btk catalytic activity (*Rawlings et al., 1996*). Phosphorylation of Tyr 223 has been reported to have no apparent influence on Btk catalytic activity (*Park et al., 1996*; *Middendorp et al., 2003*). It lies within the peptide-binding groove of the SH3 domain, and its phosphorylation alters SH3 domain selectivity among binding partners (*Morrogh et al., 1999*; *Middendorp et al., 2003*).

Binding of a Tec-kinase PH domain to the membrane lipid $PIP_3$, produced as a consequence of receptor activation, provides a simple control mechanism for switching on the kinase. Membrane recruitment triggers trans-autophosphorylation, through increased local concentration, or trans-phosphorylation by membrane-bound Src-family kinases, thereby activating the Tec kinase (*Rawlings et al., 1996*).

We present here an analysis of two crystal structures of Btk, which jointly provide molecular details of its autoinhibition. One structure is of the Src-like module, in an assembled and inactive conformation, resembling those of inactive c-Abl and of the Src kinases c-Src and Hck (*Sicheri et al., 1997*; *Xu et al., 1997*; *Nagar et al., 2003*). The other is of a Btk construct that lacks the SH3 and SH2

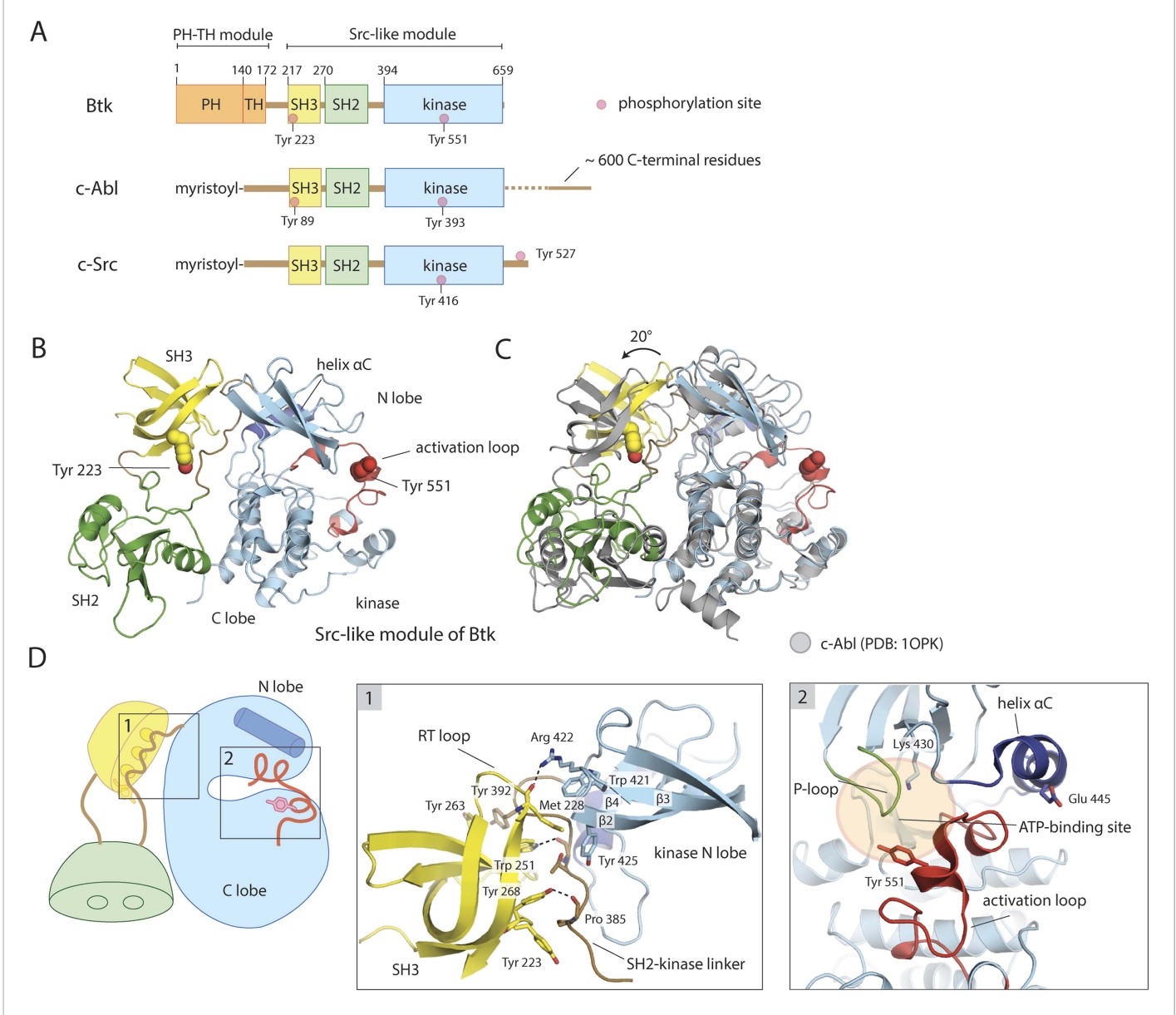

**Figure 1**. Crystal structure of the Src-like module of Btk. (**A**) Domain architectures of Btk, c-Abl, and c-Src. (**B**) Model for the Src-like module of Btk, based on the crystal structure of the domain-swapped dimer. (**C**) Comparison of the Src-like modules of Btk and c-Abl (PDB: 1OPK) (*Nagar et al., 2003*), superimposed on the C lobes of the kinase domains. There is a relative rotation of about 15˚ for the SH2 domains and 20˚ for the SH3 domains for the two structures. (**D**) Details of the SH3/SH2-kinase linker interface and the kinase catalytic cleft in Btk. Left panel: a cluster of hydrophobic residues on the SH3 domain packs against the polyproline type-II helix formed by the SH2-kinase linker. This type of interaction is seen in most SH3-peptide ligand complexes. Right panel: the activation loop of Btk folds into the mouth of the catalytic cleft, blocking part of the ATP-binding cleft. Glu 445, in helix αC, forms an ion pair with Lys 430 in the active conformation but is prevented from doing so in this inactive conformation by the activation loop and helix αC.

The following source data and figure supplements are available for figure 1:

**Source data 1**. Structures of fragments of Tec family kinases in the Protein Data Bank.

**Figure supplement 1**. Structural details of the Src-like module of Btk.

modules and that links the PH-TH module directly to the kinase domain. We find that the PH-TH module stabilizes the inactive conformation of the kinase by interacting with the N lobe. The inactive conformation of Btk is thus a compact one, in which the PH-TH module cooperates with the SH2 and SH3 domains to maintain an inactive structure. Several structures of isolated domains of Btk have been determined previously, as summarized in *Figure 1—source data 1*, which also lists structures of fragments of other Tec family kinases.

We also report a set of findings that reveal an unanticipated role for the PH-TH module in activating the kinase domain. We find that inositol hexakisphosphate ($IP_6$), a soluble signaling molecule ubiquitous in animal and plant cells (*Szwergold et al., 1987*; *Vallejo et al., 1987*), can activate Btk strongly. We have determined a crystal structure of the PH-TH module of Btk bound to $IP_6$ and identified a previously undetected binding site on the PH-TH module, distinct from the canonical $PIP_3/IP_4$ site. The new site is critical for activation by $IP_6$.

We show that Btk activation by $IP_6$ is a consequence of a transient, $IP_6$-induced PH-TH dimerization, which promotes transphosphorylation of the kinase domains. The possibility that PH dimerization might be important for Btk activation was first proposed by Saraste and colleagues (*Hyvönen and Saraste, 1997*), and we now show that the dimer interface identified previously is essential for $IP_6$-dependent Btk activation. $IP_6$-triggered transient dimerization of the PH domain may provide a membrane-independent mechanism for activating Btk in the nucleus or cytoplasm or may have a role in feedback stimulation of Btk that has been activated by $PIP_3$ at the membrane. The prior model for Btk activation involved only a $PIP_3$-dependent membrane recruitment as the key step in switching on the kinase. The ability of $IP_6$ to activate Btk points to an as yet unappreciated complexity to the activation mechanism.

## Results and Discussion

### Structural analysis of Btk

We have determined four crystal structures as part of this work. Two of these structures help explain how Btk is autoinhibited. These are the structures of the Btk SH3-SH2-kinase module (determined at 2.6 Å resolution) and of the PH-TH-kinase unit (determined at 1.7 Å resolution). The other two structures are of the isolated PH-TH module bound to $IP_6$ (determined at 2.3 Å resolution) and the isolated kinase domain with mutations in the activation loop (determined at 1.6 Å). The details of the crystallographic experiments are given in *Tables 1, 2*.

### Crystal structure of the Btk SH3-SH2-kinase module

We determined the structure of a construct of mouse Btk spanning residues 214 to 659, which starts just before the SH3 domain and ends at the C terminus of full-length Btk. The structure resembles those described previously for autoinhibited, Src-like, non-receptor tyrosine kinases, with the SH2-kinase linker sandwiched between the SH3 domain and the small domain of the kinase (the N lobe of the kinase) (*Figure 1B*). The crystal structure has two molecules of Btk in the asymmetric unit, intertwined as a domain-swapped dimer in which each SH2 domain has pieces from two molecules (*Figure 1—figure supplement 1A*). The domain-swapped SH2 domain packs against the large domain of the kinase (the C lobe of the kinase), but Btk lacks a Src-like C-terminal tail to stabilize this interaction.

The sequence identity between Btk and c-Abl is 42%, 31%, and 46% in the SH3, SH2, and kinase domains, respectively; for Btk and c-Src, it is 44%, 29%, and 40%. Superposition of the C lobes of the Btk and c-Abl kinase domains gives an approximately 20° relative rotation of their SH3 domains; superposition of the SH3 domains, a relative rotation of about 15° for their SH2 domains. Similar results are obtained for a comparison to c-Src. Coupling between the SH3 and SH2 domains may therefore be somewhat different in these kinases (*Figure 1C*).

The structure of the SH3 domain of Btk is nearly identical to those determined previously using NMR (the r.m.s. deviation for 53 Cα atoms is 0.9 Å) (*Hansson et al., 1998*; *Tzeng et al., 2000*), except for the expected differences seen at the N and C termini. When compared with the other autoinhibited SH3-SH2-kinase structures (c-Src, Hck, c-Abl), the conserved features of the SH3/linker interaction in Btk are the polyproline type-II helix conformation adopted by residues 383–387 in the SH2-kinase linker, including the proline at position 385, and the overall hydrophobic character of the peptide-binding pocket on the SH3 domain (*Figure 1D*). The tryptophan sidechain in the center of this pocket donates a hydrogen bond to a backbone carbonyl of the linker, as in most other SH3-ligand complexes. A cluster

**Table 1.** Data collection and refinement statistics

| | Src-like module of mouse Btk (217-659) | PH-TH-kinase unit of bovine Btk | Btk PH-TH module bound to $IP_6$ | Btk kinase domain with mutations in the activation loop |
|---|---|---|---|---|
| PDB ID | 4XI2 | 4Y93 | 4Y94 | 4Y95 |
| Data collection | | | | |
| | $KAu(CN)_2$ | Native | Native | Native |
| Wavelength (Å) | 0.9474 | 1.000 | 1.000 | 1.000 |
| Space group | P $3_1$ 2 1 | P 2 $2_1$ $2_1$ | P 1 | P 1 |
| $a,b,c$ (Å) | 132.2, 132.2, 107.6 | 78.6, 38.3, 157.6 | 37.2, 64.0, 80.0 | 50.9, 79.0, 79.2 |
| $\alpha,\beta,\gamma$ (°) | 90.0, 90.0, 120.0 | 90.0, 90.0, 90.0 | 82.0, 88.8, 89.8 | 90.7, 89.9, 90.0 |
| Resolution (Å) | 43.2–2.6 | 50–1.7 | 43.9–2.3 | 47.9–1.6 |
| $R_{sym}$ (%) | 8.7 (>100) | 12.2 (79.7) | 6.0 (53.8) | 5.1 (71.7) |
| $I/\sigma(I)$ | 10.6 (1.1) | 17.0 (3.5) | 10.8 (1.2) | 13.2 (3.2) |
| Completeness (%) | 91.9 (69.6) | 99.2 (98.5) | 94.9 (71.8) | 96.9 (94.7) |
| Redundancy | 4.8 (3.5) | 7.8 (7.4) | 2.0 (1.7) | 24.4 (3.1) |
| Wilson B factor | 72.8 | 15.7 | 52.3 | 18.1 |
| Refinement | | | | |
| Resolution | 43.2–2.6 | 50–1.7 | 46.5–2.3 | 47.9–1.6 |
| Reflections | 57,643(30,933) | 53,357 | 30,516 | 157,562 |
| $R_{free}$ reflections | 1525 | 2000 | 2025 | 2009 |
| $R_{work}/R_{free}$ | 0.237/0.252 | 0.167/0.195 | 0.236/0.254 | 0.155/0.187 |
| No. atoms | | | | |
| Protein | 3459 | 6970 | 5053 | 8911 |
| Ligands | 2 | 80 | 216 | 961 |
| Water | 0 | 525 | 39 | 737 |
| Average $B$ factors | | | | |
| Protein | 114.4 | 27.7 | 62.1 | 22.5 |
| Solvent | N/A | 33.0 | 66.2 | 30.1 |
| Root mean square deviation from ideality | | | | |
| Bonds (Å) | 0.006 | 0.005 | 0.003 | 0.014 |
| Angles (°) | 1.11 | 0.977 | 0.764 | 1.571 |
| Ramachandran statistics | | | | |
| Favored (%) | 91 | 98.58 | 98.2 | 97.11 |
| Disallowed (%) | 2.4 | 0.0 | 0.0 | 0.18 |
| MolProbity clash score | 9.6 | 2.8 | 5.16 | 2.66 |

The $CC_{1/2}$ values for the PH-TH-kinase dataset, $IP_6$-bound PH-TH dataset and the kinase domain with mutations in the activation loop dataset are 99.9 (86.5), 99.9 (55.4) and 99.9 (90.7), respectively.

of hydrophobic sidechains extends from the binding pocket and the SH3 'RT loop', across the SH2-kinase linker segment, to the β2-β3 loop and strand β4 of the kinase (*Figure 1D*). Binding of exogenous protein ligands to the Btk SH3 domain would disassemble the Src-like inactive conformation, as seen for the Src family kinase Hck (*Moarefi et al., 1997*).

Diffuse electron density for the SH2 domain at every stage of our structure determination shows that this part of the polypeptide chain occupies a range of positions and adopts multiple local conformations. The domain itself has a marginally stable internal structure, as shown by the domain swap, and it is probably imperfectly locked in place. The SH2 βA/αA loop lies between kinase-domain

**Table 2**. Data statistics for the Src-like module of Btk

| | **Src-like module of mouse Btk (217-659)** | | | |
|---|---|---|---|---|
| Data collection | | | | |
| | KAu(CN)$_2$ | DMA | Au2(NO$_3$)$_3$ | native |
| Wavelength (Å) | 0.9474 | 0.9474 | 0.9474 | 0.9474 |
| Space group | P3$_1$ 2 1 | P3$_1$ 2 1 | P3$_1$ 2 1 | P3$_1$ 2 1 |
| a,b,c (Å) | 132.2, 132.2, 107.6 | 132.5, 132.5, 107.3 | 131.9, 131.9, 107.6 | 131.8, 131.8, 107.0 |
| α,β,γ (°) | 90.0, 90.0, 120.0 | 90.0, 90.0, 120.0 | 90.0, 90.0, 120.0 | 90.0, 90.0, 120.0 |
| Resolution (Å) | 43.2–2.6 | 41.7–3.5 | 41.7–3.4 | 40–4.0 |
| $R_{sym}$ (%) | 8.7(>100) | 8.7 (32.7) | 7.5 (32.5) | 5.1 (71.7) |
| I/σ(I) | 10.6 (1.1) | 9.7 (3.0) | 12.1 (3.6) | 18.4 (12.3) |
| Completeness (%) | 91.9 (69.6) | 92.4 (94.2) | 94.5 (95.9) | 92.3 (100) |
| Redundancy | 4.8 (3.5) | 4.3 | 4.1 | N/A |
| Wilson B factor | 72.8 | 58.8 | 70.8 | 67.3 |

helices αB and αF. Density features for this loop are relatively sharp, and the rest of the domain may pivot somewhat about its contact with the C lobe.

The domain swap occurs in the same location within the SH2 domain as a domain swap seen in a Grb2 SH2-domain crystal structure (*Schiering et al., 2000*). The SH2 domain opens up at a position within the subsidiary β-sheet, so that helix B of one polypeptide chain packs against the main β-sheet of the other and vice versa (*Figure 1—figure supplement 1B*). This exchange occurs in solution for Btk, as only the dimeric species crystallizes. The similar domain swap seen in two SH2 domains not closely related in their specificity or functions suggest that other SH2 domains may also have the same bipartite character.

There is no evidence that a domain-swapped dimer (or, indeed, any stable dimer) is a functional state of intact Btk, and so it is likely that the assembled state involves a single polypeptide chain, rather than two inter-folded chains. The relatively high concentration of purified protein could have promoted dimer formation in our preparations. In the case of Grb2, the structure of the intact protein, a monomer in solution, shows no domain swap, and loss of contacts between the deleted N- and C-terminal SH3 domains can account for destabilization of the central SH2 when it is prepared as an isolated domain (*Maignan et al., 1995*). For Btk, the PH-TH module might stabilize the Src-like module.

The unphosphorylated activation loop of Btk folds into the mouth of the catalytic cleft, blocking part of the substrate-binding site, and helix αC lies at the margin of the N lobe, preventing formation of a Glu-Lys salt bridge conserved in all catalytically active protein kinases (*Taylor and Kornev, 2011*) (*Figure 1D*). The activation loop of Btk is in a conformation seen in the structure of a Btk kinase domain-inhibitor complex (PDB: 3OCS; [*Di Paolo et al., 2011*]), and in autoinhibited c-Src (the r.m.s. deviation from c-Src is 2.3 Å for the 24 α-carbons of the activation loop), but the cleft between the two lobes of the kinase is substantially more open in Btk than in c-Src. Helix αC is also displaced from the catalytic cleft, as in inactive c-Src or cyclin-dependent kinases. The segment of the activation loop that contacts the N lobe shifts with it, with respect to c-Src as a reference, by up to 3 Å, away from the C lobe.

A network of polar interactions present in the autoinhibited forms of c-Src and Hck is also present in the Btk conformation in our crystals (*Figure 1—figure supplement 1C*). Glu 445, which would buttress Lys 430 in the active conformation, interacts instead with Arg 544 and Tyr 545. Arg 544 stacks in turn on Arg 520, clamping the latter against the phenyl ring of Phe 574. Each of these arginines also appears to have an amino–aromatic interaction: Arg 544 with Phe 517 and Arg 520 with Phe 574. In the active conformation of Btk, the same two arginines are expected (based on the structure of the active Lck kinase domain, PDB: 3LCK; [*Yamaguchi and Hendrickson, 1996*]) to engage in salt bridges with a phosphorylated Tyr 551. Activation thus entails a shuffling of salt-bridge and polar hydrogen-bond partners, described in the case of Src kinases as an 'electrostatic switch' (*Ozkirimli and Post, 2006*; *Banavali and Roux, 2009*) (*Figure 1—figure supplement 1C*). In addition, the inactive conformation is

stabilized by interactions made by the SH2-kinase linker. Leu 390 of the linker (corresponding to Trp 260 in c-Src) is sandwiched between residues Trp 421 and Tyr 461 in the kinase domain (*Figure 1—figure supplement 1D*). The hydrophobic packing between Trp 421 and Leu 390 would not be possible in the active conformation of the kinase domain.

The structure just described demonstrates that the SH3-SH2 domain in a Tec-family member can assemble into a regulatory clamp, just as it does in other Src-like kinases. In the crystal structures of c-Src, Hck, and c-Abl, a latch is present—either in the form of a phosphorylated C-terminal tail (c-Src, Hck) or a myristoyl group bound in a pocket on the C lobe of the kinase domain (c-Abl). In both cases, the latch secures the SH2-domain/C lobe contact, precisely the interface that appears to be poorly formed in our Btk crystals. As we show below, the latch in Btk is provided by the PH-TH module.

## Structure of a PH-TH-kinase construct of Btk

We expressed a 'PH-TH-kinase' construct of bovine Btk that contains the PH-TH module connected to the kinase domain by the 13-residue SH2-kinase linker. We replaced six residues in the activation loop of Btk with the corresponding ones in Itk, shown previously to stabilize the activation loop of the Btk kinase domain (*Joseph et al., 2013*). This improved the poorly ordered crystals obtained with the unmutated form, leading to a structure of the PH-TH-kinase with a bound ATP-competitive inhibitor (CGI1746, [*Di Paolo et al., 2011*]) at 1.7 Å resolution (*Figure 2A*).

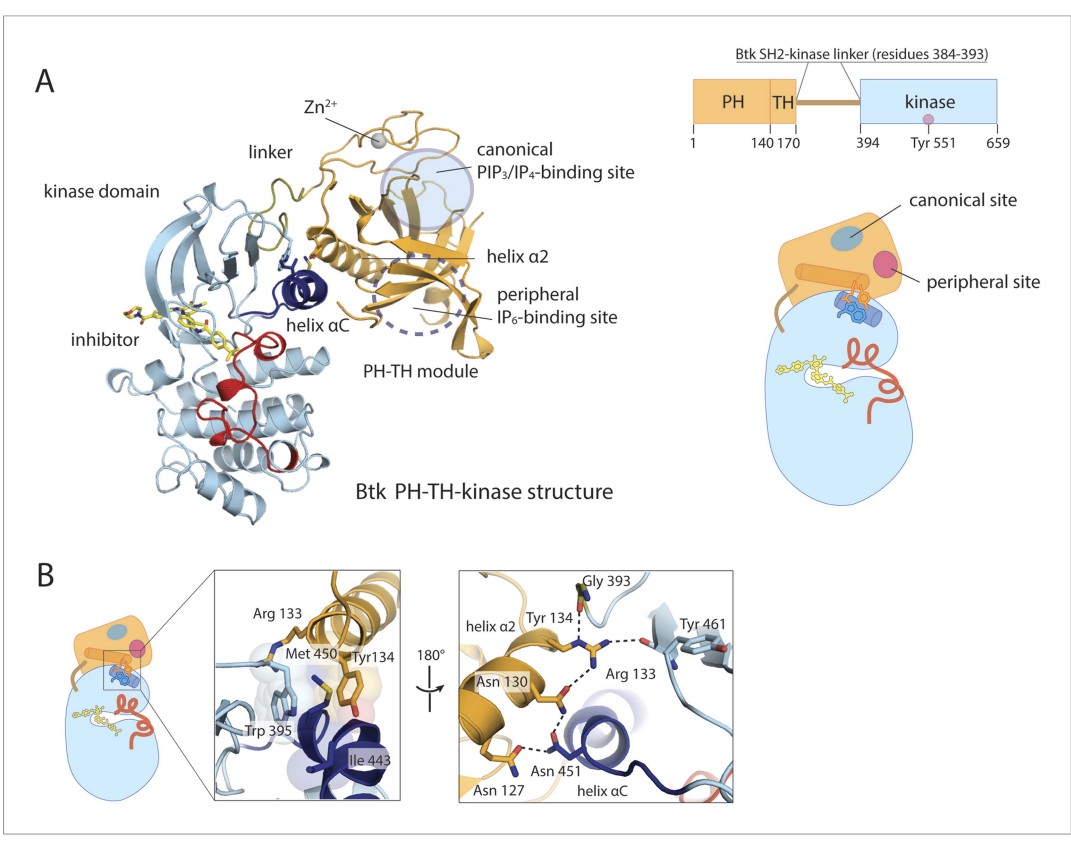

**Figure 2**. Crystal structure of the PH-TH-kinase construct of Btk. (**A**) Crystal structure of the PH-TH-kinase construct. The PH-TH module is connected to the kinase domain via a 13-residue linker from the SH2-kinase linker of Btk. The kinase domain is bound to the inhibitor CGI1746 (*Di Paolo et al., 2011*). The canonical PIP$_3$/IP$_4$ binding pocket of the PH domain is located distal to the interface with the kinase, and the peripheral binding site for IP$_6$ is indicated. (**B**) The interface between the PH-TH module and the kinase domain of Btk.

The following figure supplement is available for figure 2:

**Figure supplement 1**. Structure of the kinase domain of Btk and its interaction with the PH domain.

The overall conformation of the kinase domain, as well as the specific conformation of the activation loop, is very similar in the two structures we have determined (*Figure 2—figure supplement 1A*). We also determined the structure of the isolated Btk kinase domain with the six mutations described above and found by comparing it to wild-type Btk that the mutations do not introduce major perturbations in the structure (*Figure 2—figure supplement 1A*). Details of expression, purification, crystallization, and structure determination are in the 'Materials and methods'.

The PH-TH module contacts the kinase N lobe. The interface differs substantially from the one found in Akt1, another PH-domain regulated kinase (*Wu et al., 2010*) (*Figure 2—figure supplement 1B*). In Akt1, the PH domain sits in front of the catalytic cleft of the kinase, blocking peptide substrate binding. In Btk, the PH-kinase interface does not directly occlude the active site. The interface combines a set of hydrophobic contacts with a hydrogen-bond network (*Figure 2B*). In particular, Tyr 134 at the C terminus of helix α2 of the PH-TH packs into a groove between Trp 395 at the linker-kinase junction and helix αC in the kinase N lobe. We expect these interactions to stabilize the inactive conformation of the kinase, by hindering the shift in helix αC that accompanies kinase activation. Experiments confirming these expectations are described below, in the section on 'Autoinhibition'.

## A structural model for full-length Btk

We combined the two structures just described to build a model for full-length Btk (*Figure 3A*). Aligning the kinase domains in the two structures gives some local overlap between helix α2 in the PH-TH module and the β3/β4 loop in the SH3 domain, but otherwise compatible positions for the regulatory domains (*Figure 3—figure supplement 1A*). We therefore carried out molecular dynamics simulations of the two component structures and of the composite assembly, to determine whether energetically minor adjustments could resolve the remaining clashes. The trajectories ranged in length from 60 to 150 ns (see 'Materials and methods' for computational details).

In simulations of the SH3-SH2-kinase module, the initial and final conformations of the SH3 and kinase domains had r.m.s. deviations in Cα positions of only ~3.0 and ~2.0 Å, respectively, after first aligning the C lobe of the kinase domain (*Figure 3—figure supplement 1B*). The SH2 domain shifted by ~6 Å, with a roughly 10° rotation that brought it closer to the C lobe of the kinase domain and closer to its position in the inactive forms of c-Src, Hck, and c-Abl. Some adjustment of the SH2 domain in these simulations was expected, as the domain swap in the crystals probably perturbed the domain to some extent, and the starting structure for the simulations was therefore itself an approximation.

In simulations of the PH-TH-kinase construct, the interface with the kinase domain remained intact, but the PH-TH module wobbled back and forth through a rotation of about 20° (*Figure 3—figure supplement 1C*). The averaged r.m.s. deviation between the crystal structure and structures from the end of the trajectories, aligned on the kinase domain, was ~4 Å for the PH-TH module. The PH-TH module pivots about an apparent anchor point at the C terminus of helix α2, at the center of the autoinhibitory contact (*Figure 3—figure supplement 1C*).

The fluctuations in the PH-TH orientation generate transient structures in which there would be essentially no clashes between the SH3 domain and the PH-TH module in the merged structures. We initiated multiple MD trajectories from such a composite model (*Figure 3A*). The inhibitory interactions between each of the regulatory modules and the kinase domain were preserved, but various intra-domain rearrangements occurred during the course of these trajectories. The overall conformation of Btk at the end of the trajectories is very similar to the initial structure. The averaged r.m.s. deviations of the PH-TH module and the SH3 domain between the crystal structures and the end of trajectories, aligned on the Src-like module of the kinase, were ~4 Å and ~2.5 Å, respectively. The former is comparable to the value found in simulations of the PH-TH-kinase, but the latter is larger than in simulations of the Src-like module, probably because the SH3 domain reorients with respect to the kinase domain. The SH3 rotation results in a polar interface between SH3 and PH-TH, with good electrostatic complementarity, which preserves the interactions between the SH3 domain, the linker and the N lobe of the kinase (*Figure 4B*). In these simulations, the r.m.s. deviations of the SH2 domain and the kinase N lobe are ~2.6 Å and 2.0 Å, respectively, after 360 ns, similar to those of the Src-like module simulations. We emphasize, however, that restrictions in the time scale of the simulations allow only qualitative conclusions and do not give a high-resolution picture of the intact molecule.

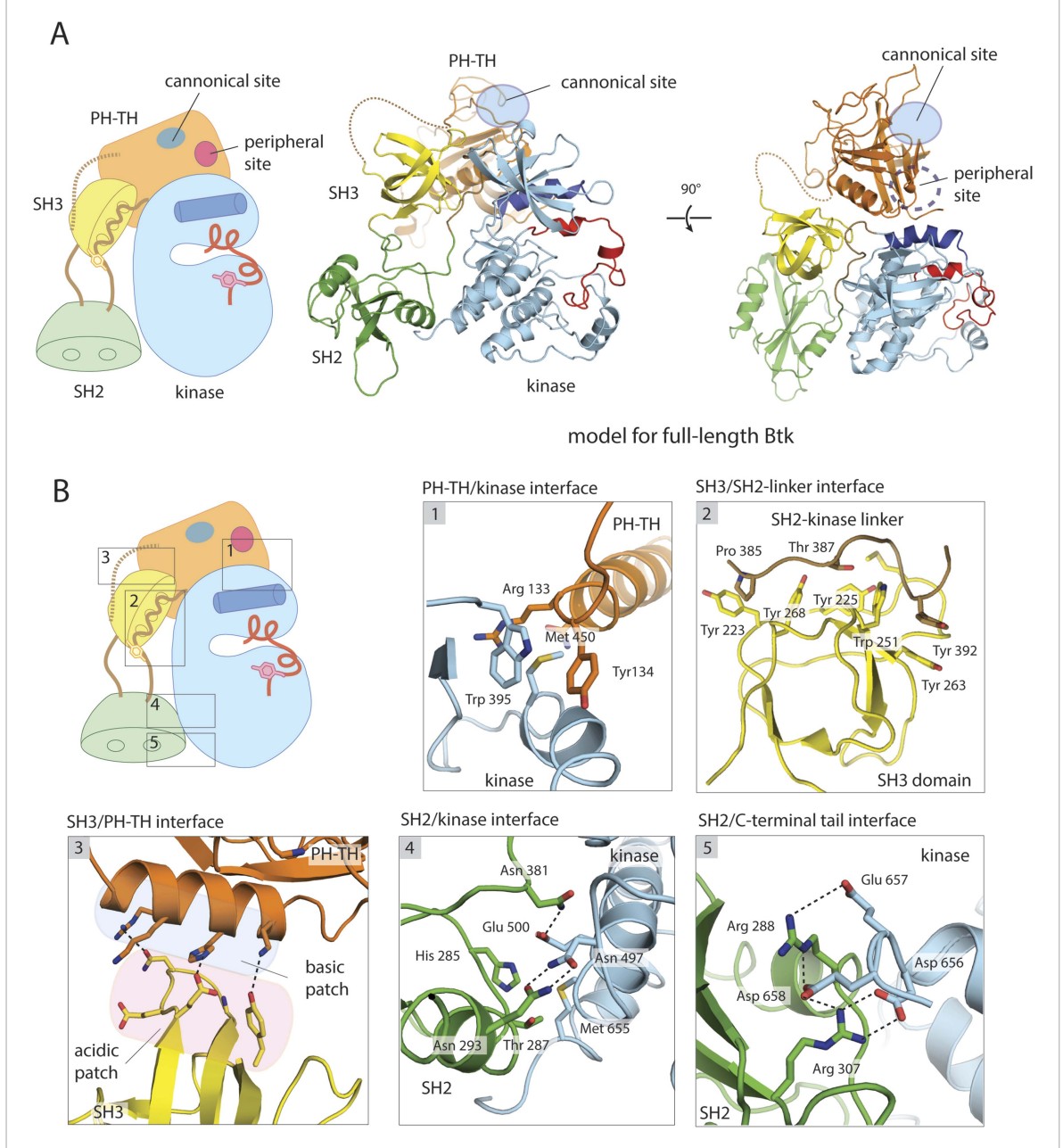

**Figure 3**. A structural model for full-length Btk. (**A**) A composite model for full-length Btk. The PH-TH domain sits on top of the kinase domain and the SH3 domain, serving as a 'latch' that presumably stabilizes the Src-like module in the autoinhibited conformation. The 45-residue linker between the PH-TH module and the SH3 domain is represented by the dotted brown line. Two orthogonal views of an instantaneous structure at 360 ns from a molecular dynamics trajectory for the composite model are shown. (**B**) Inter-domain interactions in an instantaneous structure at 360 ns from a simulation of the composite model. The SH2/kinase interface and the PH-TH /SH3 interface are formed and remain stable during the molecular dynamics simulation.

The following source data and figure supplements are available for figure 3:

**Source data 1**. Structures from several time points in the simulation of the composite model for full-length Btk.

**Figure supplement 1**. Utilization of molecular dynamics in constructing a model for full-length Btk.

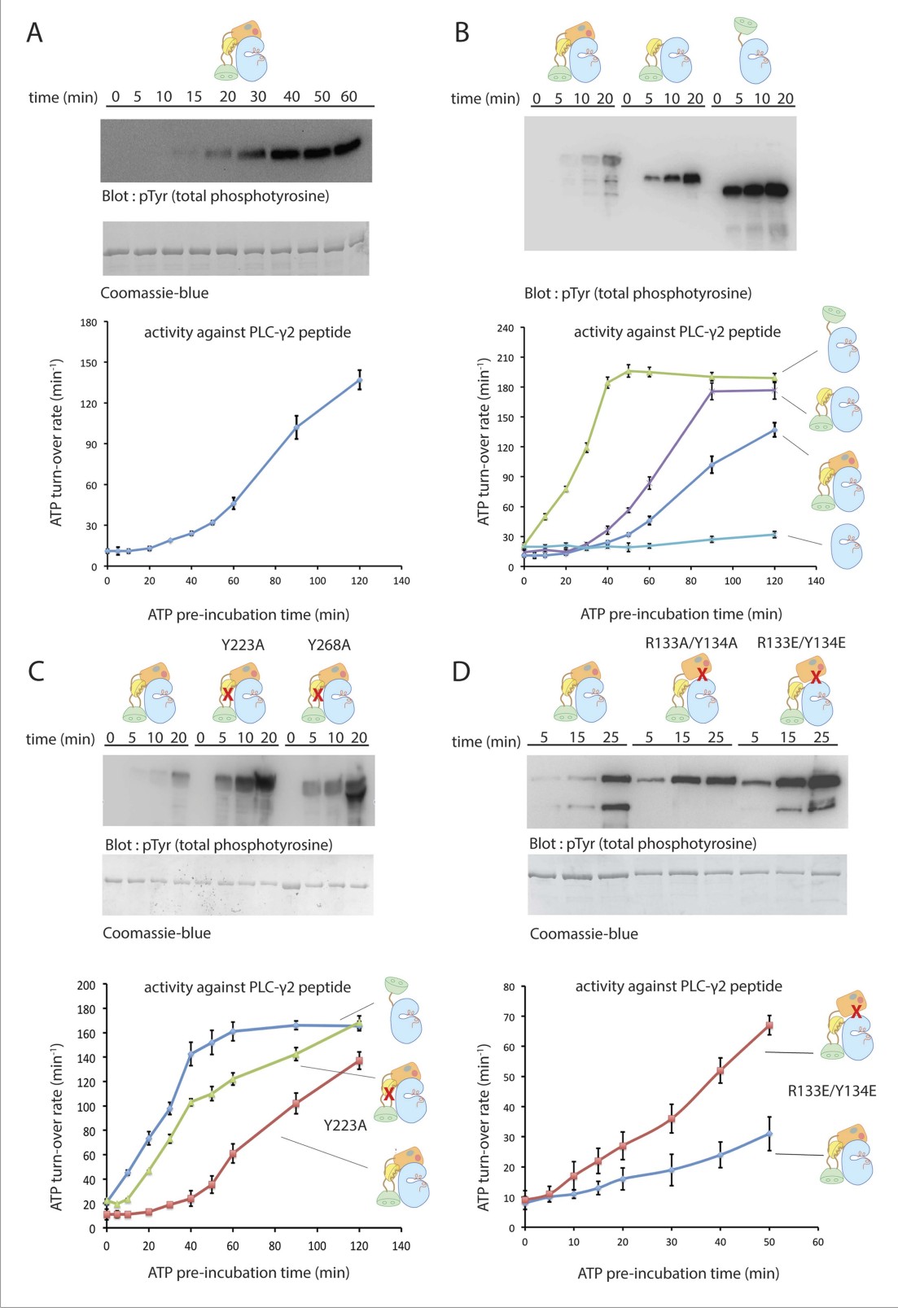

**Figure 4**. Autoinhibition of Btk. (**A**) Activation of full-length bovine Btk (residues 1 to 659, 2 µM). Reactions are carried out in the presence of 10 mM Mg²⁺, 150 mM NaCl, 1 mM ATP, 25 mM Tris-HCl pH 8.0. The level of autophosphorylation is assayed by immunoblotting an SDS-PAGE gel with a non-specific, anti-phosphotyrosine antibody (4G10, EMD Millipore) (upper panel). The amount of total protein loaded on the gel is measured by
*Figure 4. continued on next page*

*Figure 4. Continued*

coomassie-blue staining. The kinase activity of Btk is assayed by a continuous kinase-coupled colorimetric assay, in the presence of 1 mM PLC-γ2 peptide substrate. See methods for detailed experimental procedures. (**B**) Comparison of the activation of the Btk Src-like module (residues 217 to 659), SH2-kinase (residues 270 to 659), and the kinase domain (residues 394 to 659). The SH2-kinase construct activates substantially faster than full-length Btk and the Src-like module of Btk. Activated full-length Btk degrades to a small extent over time, which results in some lower molecule-weight bands being detected on the western blot. (**C**) Activation of full-length Btk with mutations Y223A and Y268A. Tyr 223 and Tyr 268 are on the SH3/SH2-linker interface, and the two mutants activate faster than wild-type Btk. (**D**) Activation of full-length Btk with a double mutation (R134E/Y133E). Arg 134 and Tyr 133 are located at the PH-TH/kinase interface.

Structures from several time points in the simulation of the composite model are deposited on the *eLife* website (*Figure 3—source data 1*).

## Autoinhibition

We probed the significance of the interactions seen in the crystal structures by studying how various mutations affect the rate of autophosphorylation and enzymatic activity. We avoided the heterogenous phosphorylation that accompanies expression in eukaryotic cells by using bovine Btk expressed in bacteria, which gives a pure, unphosphorylated product with good yield. Bovine Btk is 98.8% identical to human Btk in sequence, with only eight amino-acid differences over the entire protein. The mass of the bacterially expressed full-length Btk (76,379 Da), as determined by mass spectrometry, is consistent with the calculated molecular weight (76,381.2 Da).

Incubation with ATP-Mg$^{2+}$ initiates autophosphorylation, leading in turn to increased catalytic activity. We monitored activation in two ways. First, we monitored the phosphorylation by Btk of a peptide substrate derived from PLC-γ2, using a continuous kinase-coupled colorimetric assay (*Figure 4A*). Second, we followed accumulation of tyrosine-phosphorylated Btk by immunoblotting with a non-specific, anti-phosphotyrosine antibody (4G10, EMD Millipore) (*Figure 4A*). The results of the two assays are in good agreement.

The kinase domain of Btk has low catalytic activity and autophosphorylates very slowly, like the c-Abl kinase domain and unlike those of the Src family (*Figure 4B*). For example, there is no detectable change over 90 min in the level of phosphorylation of the Btk kinase domain at 4 µM concentration. Inclusion of the SH2 domain and the SH2-kinase linker (but not the SH3 domain or the PH-TH module) increases Btk autophosphorylation substantially (*Figure 4B*). We have not studied how the SH2 domain increases activity, but we note that in c-Abl, the SH2 domain docks onto the N lobe of the kinase domain and stabilizes the active conformation and that activation by the SH2 domain is also seen in Csk (*Sondhi and Cole, 1999*) and c-Fes (*Nagar et al., 2006*; *Filippakopoulos et al., 2008*). The apparent linear array of domains detected by small-angle x-ray scattering from a partially phosphorylated form of full-length Btk might represent the activated rather than the inactive form, in a conformation similar to that of activated c-Abl (*Márquez et al., 2003*; *Nagar et al., 2006*).

The autoactivation rate of the complete Src-like module of Btk is lower than that of the SH2-kinase module (*Figure 4B*), as expected from the joint clamping effect of the SH3 and SH2 domains. Based on contacts seen in the crystal structure of the Src-like module of Btk, we introduced (separately) two mutations, Y223A and Y268A, into full-length Btk. These SH3-domain residues pack against Pro 385 in the SH2-kinase linker, and their phosphorylation (or mutation to alanine) would destabilize the autoinhibited conformation (*Figure 1C*). The autoactivation rates of both mutants are indeed about sixfold greater than that of the unmutated enzyme and comparable to that of the SH2-kinase module (the rate of activation is estimated from the slope of activity vs time, and is only a rough measure because of non-linearity; *Figure 4C*). An analogous interaction regulates c-Abl activation: phosphorylation of Tyr 89, in the SH3-kinase interface, is necessary for full activation (*Brasher and Van Etten, 2000*).

Mutations expected to disrupt the PH-TH-kinase interface likewise increase the autoactivation rate of full-length Btk, consistent with our inferences from examining the interface between the PH-TH and the kinase N lobe (*Figure 4D*). Removal of the entire PH-TH module, while retaining the Src-like regulatory components, increases the rate of autoactivation about threefold, as estimated from the slopes of the activity curves in *Figure 4*.

## Activation of Btk by membrane recruitment

We studied the effect on Btk activity of $PIP_3$ in small unilamellar vesicles (SUVs). SUVs containing 75% DOPC, 20% DOPS, and 5% $PIP_3$ produce a rapid and substantial activation of Btk (*Figure 5A*). At a Btk concentration of 2 μM, robust phosphorylation can be detected within the first 5 min of reaction in

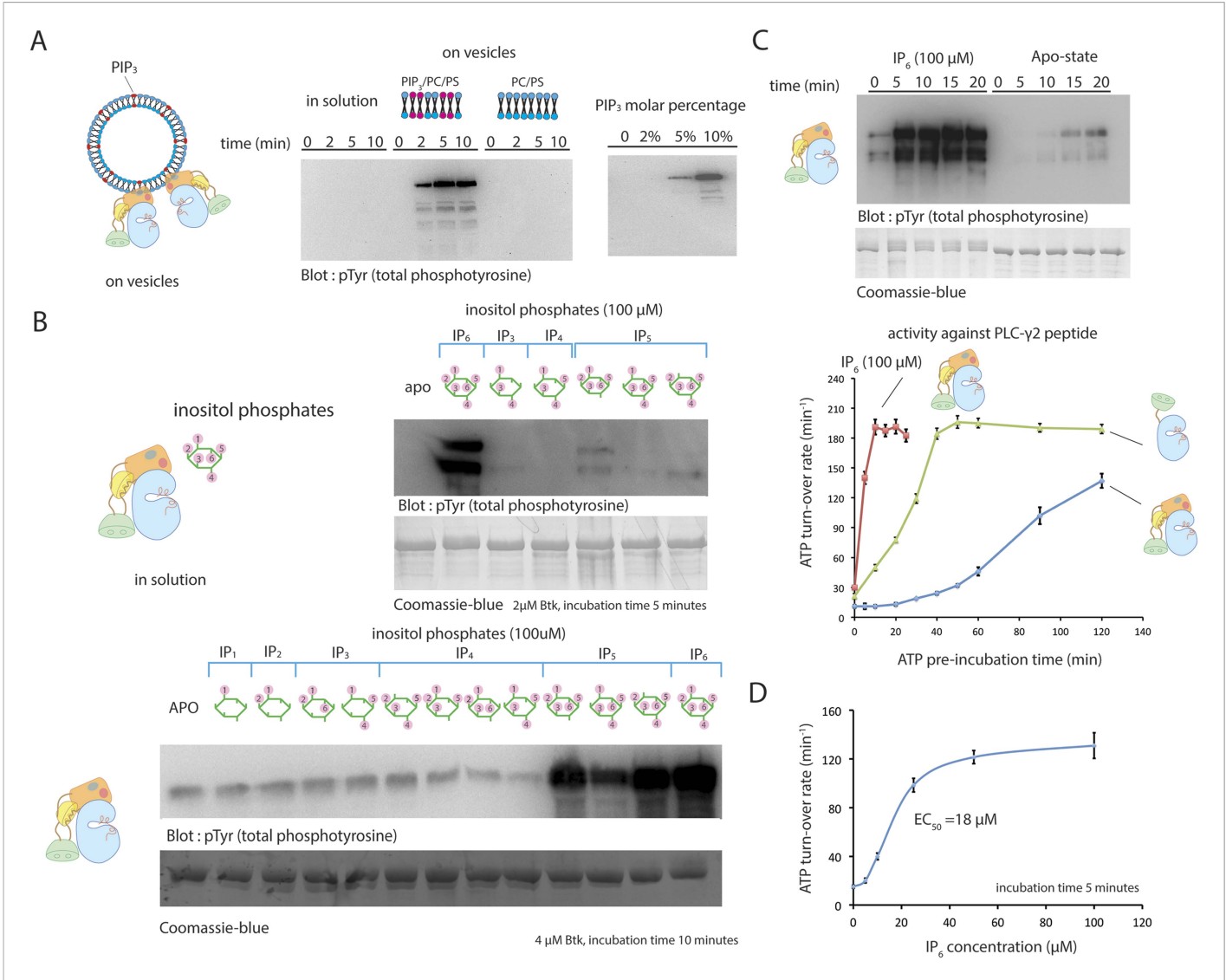

**Figure 5**. Activation of Btk. (**A**) Activation of full-length Btk on membranes. Left panel: Btk (1 μM) autophosphorylates rapidly on lipid vesicles (total lipid concentration, 1 mM) containing 75% DOPC, 20% DOPS and 5% $PIP_3$. Right panel: end-point autophosphorylation assay of Btk in the presence of lipid vesicles. The molar concentration of DOPC is kept at 75%, and the lipid fractions of $PIP_3$ is 0%, 2%, 5%, and 10%, with the remaining lipid fraction filled with DOPS. The measurements are performed after 2 min incubation with lipid vesicles in the presence of ATP/$Mg^{2+}$. (**B**) End-point autophosphorylation assays of Btk in solution in the presence of inositol phosphates. Upper panel: Btk (2 μM) autophosphorylates rapidly in the presence of 100 μM $IP_6$ in solution, with the results after the first 5 min of incubation shown here. Essentially no phosphorylation is detected in Btk samples incubated with $IP_3$, $IP_4$ or I(1,3,4,5,6)$P_5$, and very low phosphorylation is detected in Btk samples incubated with I(1,2,3,5,6)$IP_5$ and I(2,3,4,5,6)$IP_5$ in the same experiment. Lower panel: the assay is repeated with other inositol phosphates at a higher protein concentration (4 μM) and longer incubation time (10 min), and the gel was exposed for a longer time. There is a small amount of degraded Btk in the sample, as judged by the coomassie staining, leading to multiple bands in the western blot shown here and in panel **C**. (**C**) Activation of full-length Btk (2 μM) in the presence of $IP_6$ (100 μM). (**D**) Activation of full-length Btk (2 μM) with increasing concentrations of $IP_6$.

The following figure supplement is available for figure 5:

**Figure supplement 1**. Activation of various deletion constructs of Btk.

the presence of these vesicles. The extent of activation depends on the concentration of $PIP_3$ in the vesicles, presumably because increasing the $PIP_3$ fraction increases the surface density of Btk on the lipid vesicles and thereby increases the rate of trans-autophosphorylation (*Figure 5A*). An estimate of the surface density of $PIP_3$ on these vesicles suggests that the local concentration of Btk on the vesicle surface is in the millimolar range, that is, 100-fold greater than in bulk solution (see 'Materials and methods' for detailed calculations).

We next studied the effect of inositol phosphates, the soluble headgroups of various phosphatidylinositol phosphate lipids, on Btk activity. The purpose of these studies was to see whether interactions with the soluble headgroups might activate Btk independent of membrane localization, as is the case with Akt (*Calleja et al., 2009*; *Wu et al., 2010*). The soluble headgroup of $PIP_3$ is $IP_4$, and so this was included in our study, as were several other inositol phosphates (*Figure 5B*). $IP_4$ itself has no effect on the rate of Btk activation in solution, demonstrating that the role of $PIP_3$ is to recruit Btk to the membrane without necessarily triggering an allosteric effect.

## Activation of Btk by $IP_6$ in solution

We tested the effect of 10 other inositol phosphates on the activity of Btk (*Figure 5B*). Quite unexpectedly, we discovered that inositol hexakisphosphate ($IP_6$) promotes Btk autophosphorylation strongly at 2 µM protein concentration, substantially enhancing Btk catalytic activity within the first 5 min after addition (*Figure 5C*). We also saw a milder activation with certain inositol pentakisphosphates ($IP_5$), but these required higher protein concentration and longer times for appreciable autophosphorylation than did $IP_6$ (*Figure 5B*).

The $EC_{50}$ value for $IP_6$ activation is around 20 µM (*Figure 5D*), in the range of the physiological concentration of $IP_6$ in cells, which is 10 µM–80 µM (*Irvine and Schell, 2001*; *Shears, 2001*). In the presence of a saturating concentration of $IP_6$ (100 µM), full-length Btk is activated within the first 5 min of reaction time, in contrast to the 30 min and 120 min that are required for the activation of the SH2-kinase module and full-length Btk, respectively, in the absence of $IP_6$ or $PIP_3$-containing vesicles (*Figure 5A,C*). Once fully activated, the catalytic rates of various Btk constructs are the same with or without $IP_6$ addition, suggesting that the effect of $IP_6$ is to change the rate of autophosphorylation, and that it does not affect the catalytic machinery directly (*Figure 5C*).

By comparing constructs, we determined which domains in Btk are responsible for the $IP_6$ response. Activation by $IP_6$ required only the PH-TH module and not the SH3 and SH2 domains (*Figure 5—figure supplement 1A,B*). A PH-TH-kinase construct that contained the PH-TH-SH3 linker and the SH2-kinase linker (with a total linker length of 45 residues) had the same response to $IP_6$ as did full-length Btk (*Figure 5—figure supplement 1B*). PH-TH-kinase constructs with shorter linkers also responded to $IP_6$. The shortest linker necessary for activation contained only 13 residues from the SH2-kinase linker; we used this construct to determine the crystal structure described earlier (*Figure 5—figure supplement 1C*). The catalytic activity of the fully activated PH-TH-kinase construct was about threefold lower than that of full-length Btk. This lower activity can be explained by the absence of the SH2 domain.

$IP_6$ has no effect on activation of the isolated kinase domain of Btk, suggesting that it is not a direct allosteric activator of the enzyme. The protein kinase CK2 is activated by $IP_6$, which binds to the kinase domain and dislodges an inhibitory interaction between CK2 and the protein Nopp140 (*Lee et al., 2013*). It seems unlikely that $IP_6$ works by displacing an inhibitory interaction between the kinase and the PH-TH module, because the Src-like module (lacking the PH-TH module) activates only slightly faster than full-length Btk in the absence of $IP_6$. Nevertheless, in order to test this possibility, we introduced two mutations (Y133E and R134E) at the PH-TH-kinase interface that are expected to disrupt the inhibitory docking of the PH-TH module on the kinase domain. Full-length Btk bearing these two mutations retains activation by $IP_6$, showing that $IP_6$ does not act by dislodging the interactions seen in the crystal structure (*Figure 5—figure supplement 1D*).

## The canonical lipid-binding pocket of the PH-TH module is not critical for $IP_6$-mediated Btk activation

The structure of the PH-TH-kinase construct does not provide direct clues about how $IP_6$ might activate Btk. The $IP_4$-binding site on the PH-TH module (the 'canonical binding site') is located ~15 Å from the kinase-interacting surface of the PH-TH module (*Figure 2A*). The structure of the PH-TH-kinase construct, in which no inositol phosphates are bound, resembles closely that of the liganded PH domain

(*Baraldi et al., 1999*) in the vicinity of the PH-kinase interface, consistent with the failure of IP$_4$ to activate Btk.

We used isothermal titration calorimetry (ITC) to study the interaction between IP$_6$ and various constructs of Btk (*Figure 6A*). The ITC measurements show that among the four domains in Btk, only the PH-TH module interacts with IP$_6$. At 150 mM NaCl concentration, the titration curve for IP$_6$ binding to the PH-TH module can be fit by a one-site model for binding, with a dissociation constant,

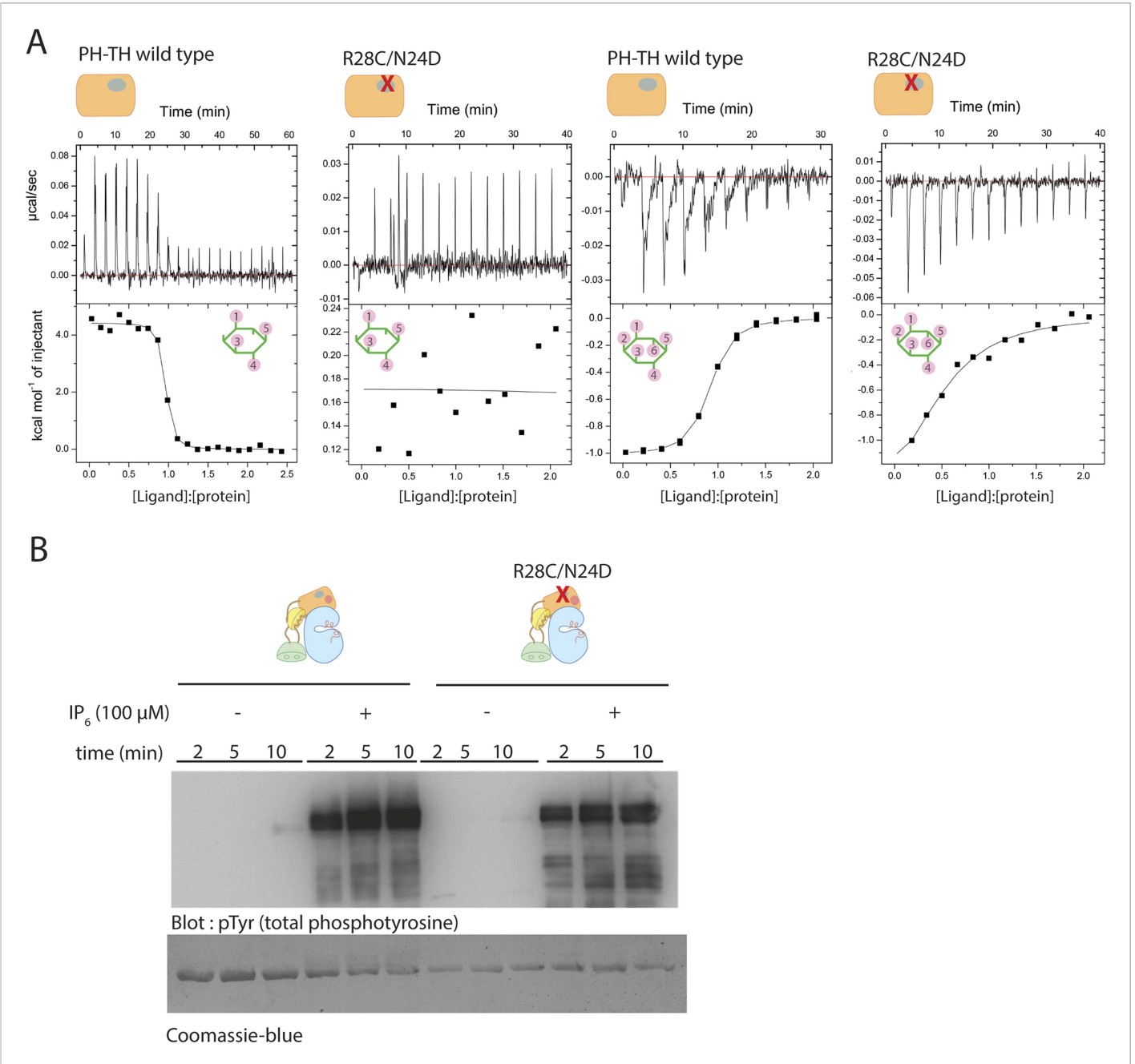

Figure 6. Binding of IP$_4$ and IP$_6$ to the Btk PH-TH module. (A) Representative isothermal titration calorimetry data for the Btk PH-TH module and its R28C/N24D variant binding to IP$_4$ or IP$_6$. The protein concentration is 20 µM and that of the inositol phosphates goes up to 300 µM. Experiments are performed at 20°C. Titration curves are shown with the baseline-corrected raw data. The parameters from fitting a one-site binding model are listed in *Table 3*. (B) Activation of full-length Btk N24D/R28C mutant (2 µM) in the presence and absence of IP$_6$ (100 µM). Asn 24 and Arg 28 are critical residues for PIP$_3$/IP$_4$ binding in the canonical lipid-binding pocket of the PH-TH module.

$K_d$, of 250 nM (*Table 3*). We introduced two mutations, R28C and N24E, at the canonical binding site in the PH-TH module. Arg 28 and Asn 24 have been shown previously to be critical for $IP_4$ binding (*Hyvönen and Saraste, 1997*; *Baraldi et al., 1999*). $IP_4$ binds to the PH-TH module with a $K_d$ value of 30 nM, and the R28C/N24E mutation in the PH-TH module reduces $IP_4$ binding to an undetectable level (*Table 3*). The ITC signal of $IP_6$ binding to the R28C/N24E mutant is reduced greatly but not completely abolished, and the residual signal corresponds to a dissociation constant between 5 μM and 10 μM.

Although the residual binding of $IP_6$ to the $IP_4$-binding deficient mutant is relatively weak, the observed affinity is consistent with the $EC_{50}$ value of 20 μM for $IP_6$ activation, as determined in our activation assays (*Figure 5C*). We observed no significant activation when the $IP_6$ concentration was lower than 5 μM, a level still high enough to saturate the canonical binding site. We also found that the R28C/N24E mutant that is deficient in $IP_4$ binding remains highly responsive to $IP_6$ stimulation, with only a slight reduction in the activation kinetics as compared to the wild-type protein (*Figure 6B*). These observations imply that the canonical binding site of the PH-TH module may not play a major role in the $IP_6$-mediated activation of Btk.

## Structure of the $IP_6$-bound PH-TH module

We determined a crystal structure of the Btk PH-TH module bound to $IP_6$ at 2.3 Å (*Figure 7*). There are four PH-TH modules in the asymmetric unit of the crystal, which form two dimers, similar to those seen in other structures of the Btk PH-TH module (*Figure 7D*) (*Hyvönen and Saraste, 1997*; *Baraldi et al., 1999*). The only structural changes in the $IP_6$-bound PH-TH module, relative to the apo- or $IP_4$-bound PH-TH module, are in the loop regions (*Figure 7—figure supplement 1B*).

The four PH-TH modules in the asymmetric unit have six bound $IP_6$ molecules, each with clearly resolved electron density. Four of the $IP_6$ molecules are in the canonical binding sites on each module. At those sites, residues Asn 24, Tyr 39, Arg 28, and Lys 12 coordinate phosphate groups on the 2, 3, and 4 positions of the myo-inositol ring; these are also the key residues involved in $IP_4$ binding (*Baraldi et al., 1999*) (*Figure 7B*). The loop between strands β1 and β2, which is important for $IP_4$ binding, is largely disordered in our structure (*Figure 7A*). There are fewer interactions at the canonical site between the PH-TH module and $IP_6$ than seen for $IP_4$, consistent with the relative affinities of the two ligands (*Figure 6A*).

The two other $IP_6$ molecules in the crystallographic asymmetric unit are bound at a peripheral site. This site is on the surface formed by an elongated β hairpin involving strands β3 and β4 of each PH-TH module. This site has not been seen in any other PH domain (*Lemmon, 2008*); it differs from the two well-characterized lipid-binding sites in the PH domains of Pleckstrin (*Ferguson et al., 1995*) and β-spectrin (*Hyvönen et al., 1995*). As in the canonical binding site, interactions between $IP_6$ and the peripheral site are predominantly electrostatistic (*Figure 7B*). Unlike the canonical site, which has a well-defined groove within which the inositol phosphates are bound, the peripheral site is relatively flat (*Figure 7B*).

A feature of the crystallographic lattice is that $IP_6$ bound at the peripheral site is sandwiched between two PH-TH modules, interacting with the peripheral sites on both (*Figure 7—figure supplement 1A,C*). The majority of interactions are with one subunit. Residues Arg 49, Lys 52, Lys 36, and Tyr 40 at the peripheral binding site of one subunit (*Figure 7—figure supplement 1C*) coordinate phosphate groups at the 1, 2, 3, and 4 positions of the myo-inositol ring. A subset of these residues, Lys 52 and Lys 36, on the other PH-TH module interact from the opposite side of the myo-inositol ring with the phosphate group at position 3 (*Figure 7—figure supplement 1C*). The bridging

**Table 3.** Thermodynamic parameters for Btk PH-TH module binding to $IP_6$ and $IP_4$

| Protein | Ligand | N | $K_a$ (×10⁶) | $K_d$ (nM) | ΔH (Kcal/mol) | ΔS (cal/mol/deg) |
|---|---|---|---|---|---|---|
| Wild-type PH-TH | $IP_6$ | 1.1 ± 0.1 | 4.2 ± 0.3 | 238 ± 32 | −1.1 ± 0.2 | 26.7 |
| Wild-type PHTH | $IP_4$ | 0.9 ± 0.1 | 39 ± 11 | 26 ± 6 | 4.4 ± 0.1 | 49.5 |
| PH-TH R28C/D24N | $IP_6$ | 0.7 ± 0.2 | 0.21 ± 0.03 | 4760 ± 700 | −1.7 ± 0.6 | 17.5 |

The integrated heat from representative isothermal titration calorimetry experiments was fit with a one-binding site model. See methods for the details of data analysis.

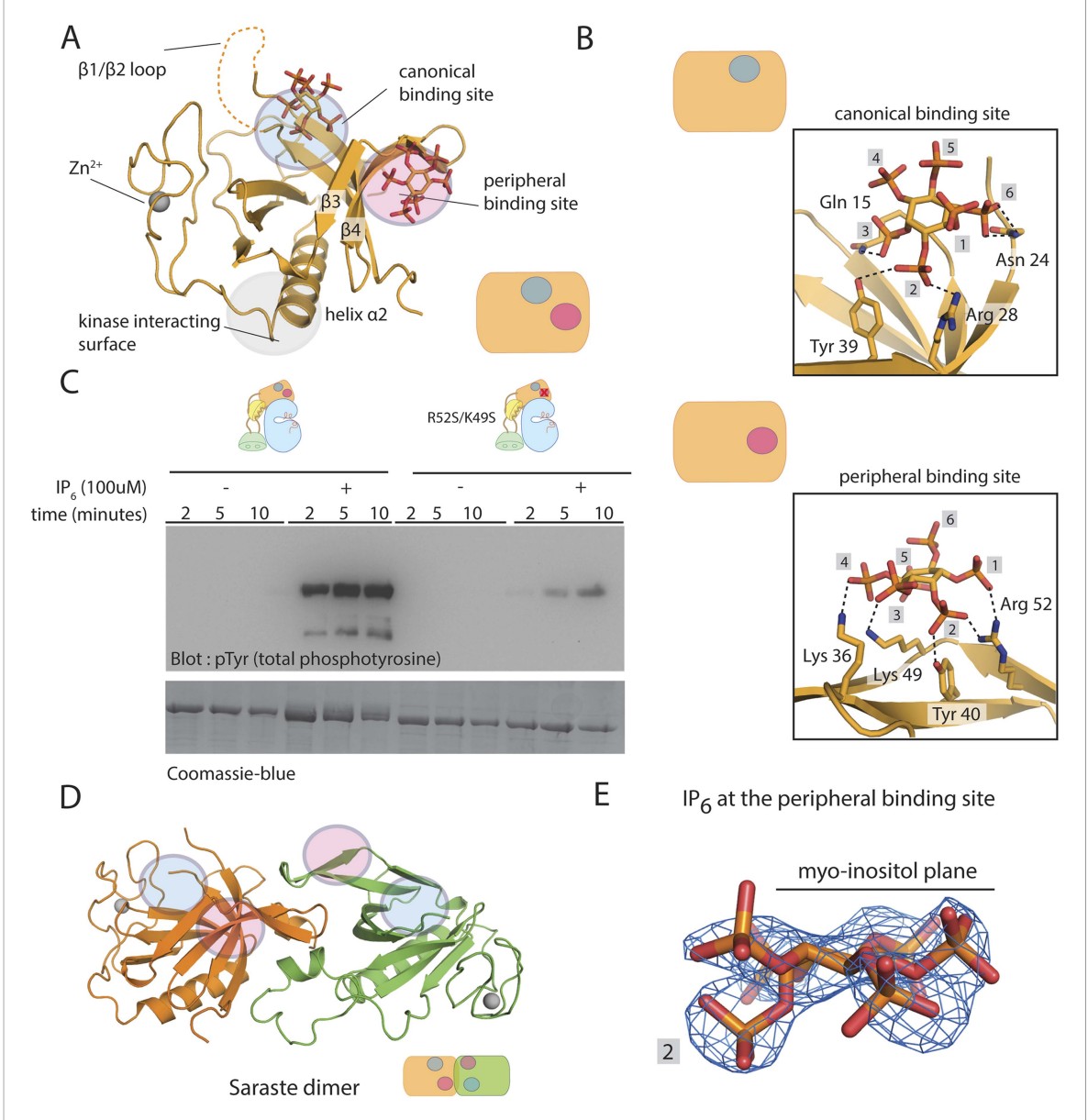

**Figure 7**. Crystal structure of the PH-TH module bound to IP$_6$. (**A**) Structure of the PH-TH module bound to two IP$_6$ molecules. One IP$_6$ molecule is in the canonical lipid-binding site and the other IP$_6$ molecule is in the peripheral binding site formed by strands β3 and β4. The β1/β2 loop (dotted lines) is disordered in this structure. (**B**) IP$_6$ coordination in the canonical binding site and in the peripheral binding site. (**C**) Activation of the Btk R52S/K49S mutant (2 μM) in the presence and absence of IP$_6$ (100 μM). Arg 52 and Lys 49 are two residues interacting with IP$_6$ in the peripheral binding site. (**D**) A dimer assembly of the Btk PH-TH module in the crystal structure of the IP$_6$-bound PH-TH module. The canonical binding site and the peripheral binding site are shown in blue and magenta circle, respectively. We refer to this dimer as the 'Saraste dimer' (*Hyvönen and Saraste, 1997*). (**E**) An electron density map, using σ$_A$-weighted 2m|F$_o$| − D|F$_c$| coefficients (*Read, 1986*) for IP$_6$ at the peripheral binding site, contoured at 1.5 σ.

The following figure supplement is available for figure 7:

**Figure supplement 1**. Aspects of the binding of IP$_6$ to Btk.

interactions provided by IP$_6$ generate an open-ended chain of PH-TH dimers in the crystal lattice (*Figure 7—figure supplement 1A*). Based on the limited interactions across this crystallographic interface, we do not believe that this open-ended chain is relevant to the function of Btk.

## The peripheral binding site is critical for IP$_6$-mediated Btk activation

We mutated Arg 49 and Lys 52, in the peripheral binding site of the PH-TH module, to serine in full-length Btk. This Btk variant autophosphorylates slowly, as does wild-type Btk, and it has very little response to IP$_6$ (*Figure 7C*). In ITC experiments, we saw no detectable binding when we added IP$_6$ to a PH-TH module with mutations at both the canonical and the peripheral sites, further confirming that the residual binding of IP$_6$ to the IP$_4$-binding deficient mutant is at the peripheral binding site (*Figure 7—figure supplement 1E*).

The disposition of the IP$_6$ molecule can be determined unambiguously from the electron density because the phosphate group at the 2 position juts out from the plane of the myo-inositol ring (*Figure 7E*). IP$_6$ at the peripheral binding site interacts with the primary PH-TH module to which it is bound through the phosphates at positions 1, 2, 3, and 4. None of the inositol phosphates used in our experiments, other than IP$_6$, can fully satisfy this criterion, which may underlie the specificity of the PH-TH module for IP$_6$. Two isomers of IP$_5$, I(2,3,4,5,6)P$_5$ and I(1,2,3,5,6)P$_5$, have a mildly activating effect on Btk; they are more effective than I(1,3,4,5,6)P$_5$. These comparisons indicate an important role in Btk activation for the phosphate at position 2 (*Figure 5B*).

In the cell, the only known enzyme that can add a phosphate to the hydroxyl group at position 2 of a myo-inositol ring is inositol 1,3,4,5,6-pentakisphosphate 2-kinase (*York et al., 1999*), which produces IP$_6$. As a consequence, IP$_6$, and inositol pyrophosphates, such as IP$_7$ and IP$_8$, may be the only cellular phosphates that can interact with the peripheral site in the Btk PH-TH module.

## A dimer of the PH-TH module is critical for IP$_6$-mediated Btk activation

Strong activation of Btk by concentration on PIP$_3$-containing membranes suggests that IP$_6$ might promote dimerization or oligomerization of Btk in solution. Neither multi-angle light scattering nor size exclusion chromatography gave convincing evidence for IP$_6$-induced association, at Btk concentrations up to 30 μM. Nevertheless, we found it striking that in every crystal structure of the Btk PH-TH module, as well as in our structure of the PH-TH-kinase construct, the module forms very similar crystallographic dimers (*Figure 7D*) (PDB entry: 1BTK [*Hyvönen and Saraste, 1997*]; PDB entry:1B55 and 1BWN [*Baraldi et al., 1999*]; see also PDB entry: 2Z0P). This dimer was first identified and discussed by Marko Hyvönen and the late Matti Saraste; we refer to it as the 'Saraste dimer'.

The Saraste dimer is formed by symmetric interactions between helix α1 and strands β3 and β4 of the PH-TH module (*Figure 7A*). The largely hydrophobic dimer interface has a buried surface area of ~1000 Å$^2$ on each molecule. Phe 44 and Tyr 42 on strand β3, Ile 9 on strand β1 and Ile 95 on helix α1 create a hydrophobic patch in one subunit of the dimer that packs tightly against the corresponding patch on the other subunit (*Figure 8A*).

To test whether the Saraste dimer interface is important for IP$_6$-mediated Btk activation, we made three mutants in which we replaced hydrophobic residues at the Saraste dimer interface with arginines and compared their autophosphorylation rates and their response to IP$_6$ with those of wild-type Btk. All three sets of mutant proteins showed little or no response to IP$_6$ (*Figure 8B*). These mutations do not affect the autophosphorylation rate of Btk in the absence of IP$_6$. We conclude that the Saraste dimer interface promotes Btk association in the presence of IP$_6$.

The activation of Class I Histone Deacetylases (HDACs) requires the formation of a multi-subunit co-repressor complex, in which Ins(1,4,5,6)P$_4$ serves as an intermolecular glue that cements the complex together (*Watson et al., 2012*; *Millard et al., 2013*). The mechanism of IP$_6$-induced transient Btk dimerization is fundamentally different, in that the peripheral IP$_6$ binding site is just outside the Saraste dimer interface, and IP$_6$ does not participate in any interactions between the two Btk subunits in that dimer.

## Fusing the Btk PH-TH module to the catalytic module of c-Abl promotes autophosphorylation of c-Abl in the presence of IP$_6$

If the effect of IP$_6$ arises from its ability to promote dimerization of the PH-TH module, fusion of a Btk PH-TH module with the catalytic domain of another kinase that autophosphorylates slowly should impart activation by IP$_6$. A good example of such a kinase is c-Abl. Indeed, in chronic myelogenous leukemia, a chromosomal translocation results in fusion of a coiled-coil oligomerization domain with c-Abl, leading to hyper-activation of the BCR-Abl fusion protein by autophosphorylation. Several other fusion variants involving c-Abl, in which different oligomerization domains are present, are associated with hematological malignancies (*De Braekeleer et al., 2011*).

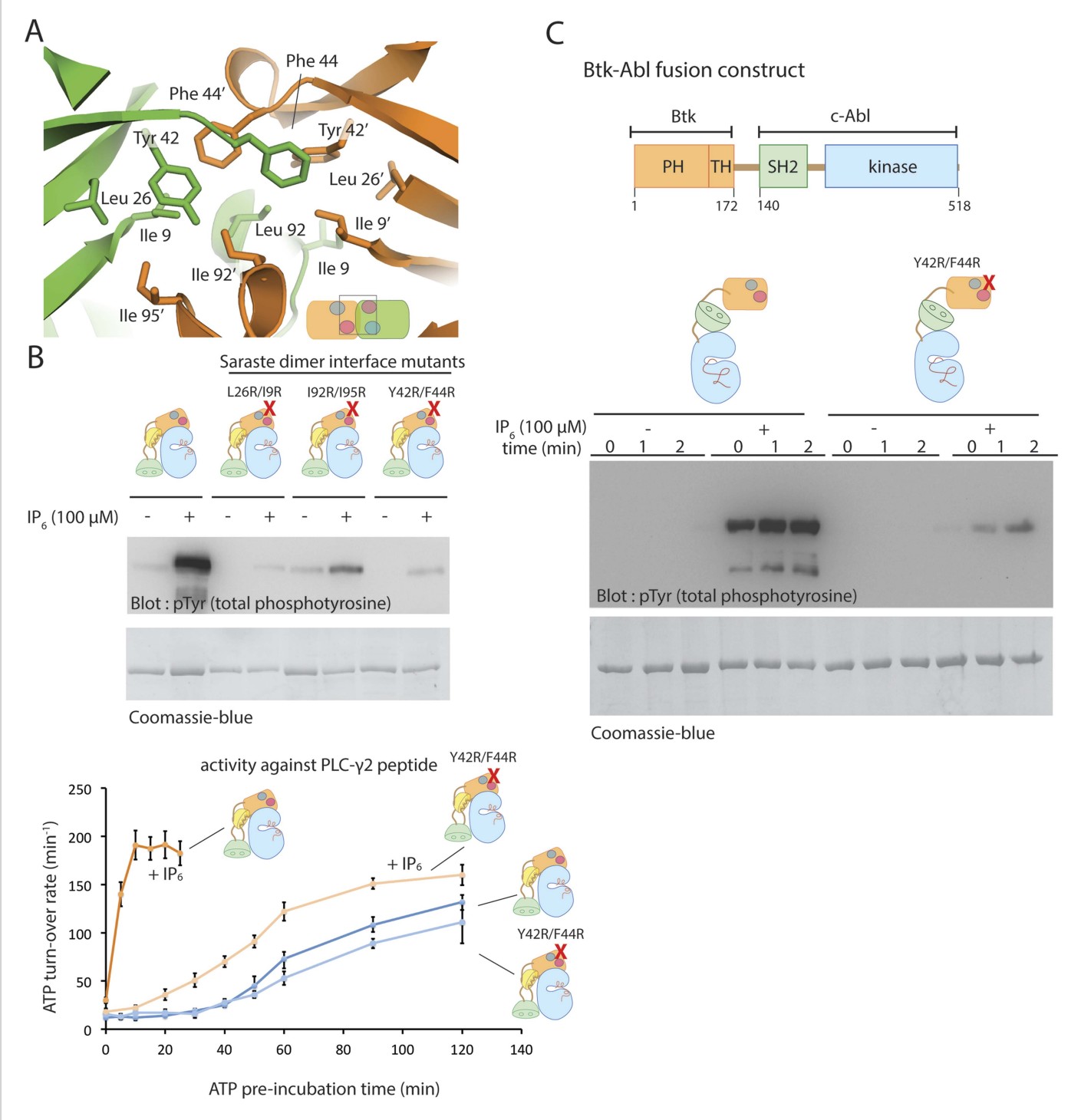

**Figure 8**. The Saraste dimer of the PH-TH module is critical for Btk activation by IP$_6$. (**A**) Molecular details of the Saraste dimer interface in the crystal structure of the IP$_6$-bound PH-TH module. (**B**) End-point autophosphorylation assays for wild-type Btk, Btk L29R/I9R mutant, Btk I94R/I95R mutant, and Btk Y42R/F44R mutant in the presence and absence of IP$_6$ (100 μM). The measurements were made after incubating the proteins with and without IP$_6$ in the presence of ATP/Mg$^{2+}$ for 5 min. Residues Leu 29, Ile 9, Ile 94, Ile 95 Tyr 42 and Phe 44 are located in the Saraste dimer interface, as shown in panel **A**. Activation of the Btk Y42R/F44R mutant is shown in the lower panel. (**C**) Activation of the Btk-Abl fusion construct (1 μM) and that of a variant of this fusion protein in which the Btk PH-TH module is mutated (Y42R/F44R). Measurements were made with 1 μM protein in the presence and absence of IP$_6$ (100 μM).

We generated a PH-TH-Abl fusion construct in which the PH-TH module of Btk is fused to the SH2-kinase module of c-Abl. We compared autophosphorylation rates of the PH-TH-Abl fusion protein with and without $IP_6$ at a saturating concentration (100 μM). The fusion protein at a concentration of 1 μM autophosphorylated slowly in the absence of $IP_6$, while the $IP_6$-bound protein became heavily phosphorylated within the first 30 s, the dead-time of our assay (*Figure 8C*). We replaced Phe 44 and Tyr 42, at the Saraste dimer interface, with arginine in the fusion protein. The autophosphorylation rate of the PH-TH-Abl mutant in the presence of $IP_6$ was much lower than that of the $IP_6$-bound wild-type protein, further evidence for the importance of the Saraste dimer interface in $IP_6$-mediated activation (*Figure 8C*).

The Btk and c-Abl kinase domains are 46% identical in sequence, but most of the surface residues are different. We infer that, except for its enzymatic activity, the interaction properties of the Btk kinase domain are not important for the $IP_6$ response. Because $IP_6$ activates both kinases when they are linked to the Btk PH-TH module, we propose that $IP_6$ promotes transphosphorylation of the attached kinase by inducing transient dimerization.

Our preliminary studies indicate that the $IP_6$-induced dimerization of the Btk PH-TH module is very weak, with a dissociation constant that is likely to be greater than 100 μM. It might appear counter-intuitive that $IP_6$ can markedly accelerate activation at 2 μM Btk concentration, well below the dissociation constant. A simple calculation shows that even a modest increase in a small dimer population can lead to a sharp change in the rate of activation because the reaction is autocatalytic. The initial production of phosphorylated Btk drives the further production of this species at an accelerating rate (*Figure 9—figure supplement 1B*).

## Speculations about the mechanism of $IP_6$ activation

Our failure to detect stable dimers of the Btk PH-TH module or of full-length Btk in the presence of $IP_6$ suggests that the monomeric PH-TH module has a conformation different from the one seen in crystal structures, which all show the same dimer contact. The high protein concentration in the crystal lattice (~30 mM) presumably drives the formation of the Saraste dimer, which is otherwise not favored.

Molecular dynamics simulations suggest that the conformation of the PH-TH module seen in the Saraste dimer is unstable when the protein is monomeric. We initiated molecular dynamics trajectories using a monomeric PH-TH module from a crystal structure which contained neither $IP_6$ nor $IP_4$. Within the first 30 ns of simulation, the β3-β4 hairpin bent towards the core of the PH-TH module (*Figure 9A* and *Figure 9—figure supplement 1A*), and Phe 44, at the heart of the Saraste dimer interface, changed its conformation and packed against the other three interfacial residues, Tyr 42, Ile 9, and Ile 95 (*Figure 9A*). This conformational change closed the dimer interface, burying the interfacial residues.

The hydrophobicity of the dimer interface drove the observed conformational change. Molecular dynamics simulations initiated from crystal structures but with the four interfacial residues substituted by alanine reduced the extent of conformational change in this area (*Figure 9A*). The peripheral binding site for $IP_6$ is adjacent to the β3-β4 hairpin that folds in and closes the dimer interface in the simulation. Thus, one effect of $IP_6$ binding may be to stabilize the open form of the hairpin.

It also appears that $IP_6$ might act by altering the electrostatic potential of the PH-TH module. We examined the surface electrostatic potential of the Btk PH-TH module by viewing graphical displays of solutions to the Poisson-Boltzmann equation (*Dolinsky et al., 2004*, *2007*). The PH-TH domain carries seven net positive charges; it is also electrostatically polarized, with the positively charged residues concentrated on the broad, contiguous surface composed of the β3-β4 hairpin, the $IP_4$ binding loop and the N-terminal region of the loop connecting strand β5 and helix α2. In the Saraste dimer, positive electrostatic field lines calculated based on the monomer run into each other at the dimer interface, suggesting that electrostatic forces oppose dimer formation.

We introduced two charge-reversal mutations at the peripheral $IP_6$-binding site (K49E and R52E), to reduce the positive charge on the protein by four units. The autophosphorylation rate of full-length Btk with the K49E/R52E mutations was substantially higher than that of the wild-type protein, with pronounced phosphorylation observed in the first 2 min of the reaction (*Figure 9B*).

We suggest that there are two barriers to the dimerization of the PH-TH module of Btk. First, the dimer interface of the PH domain is probably closed in the monomeric protein. Second, an electrostatic repulsion between PH domains further impedes dimerization. $IP_6$ binding might stabilize the open form of the PH-TH module allosterically, while also alleviating the electrostatic repulsion.

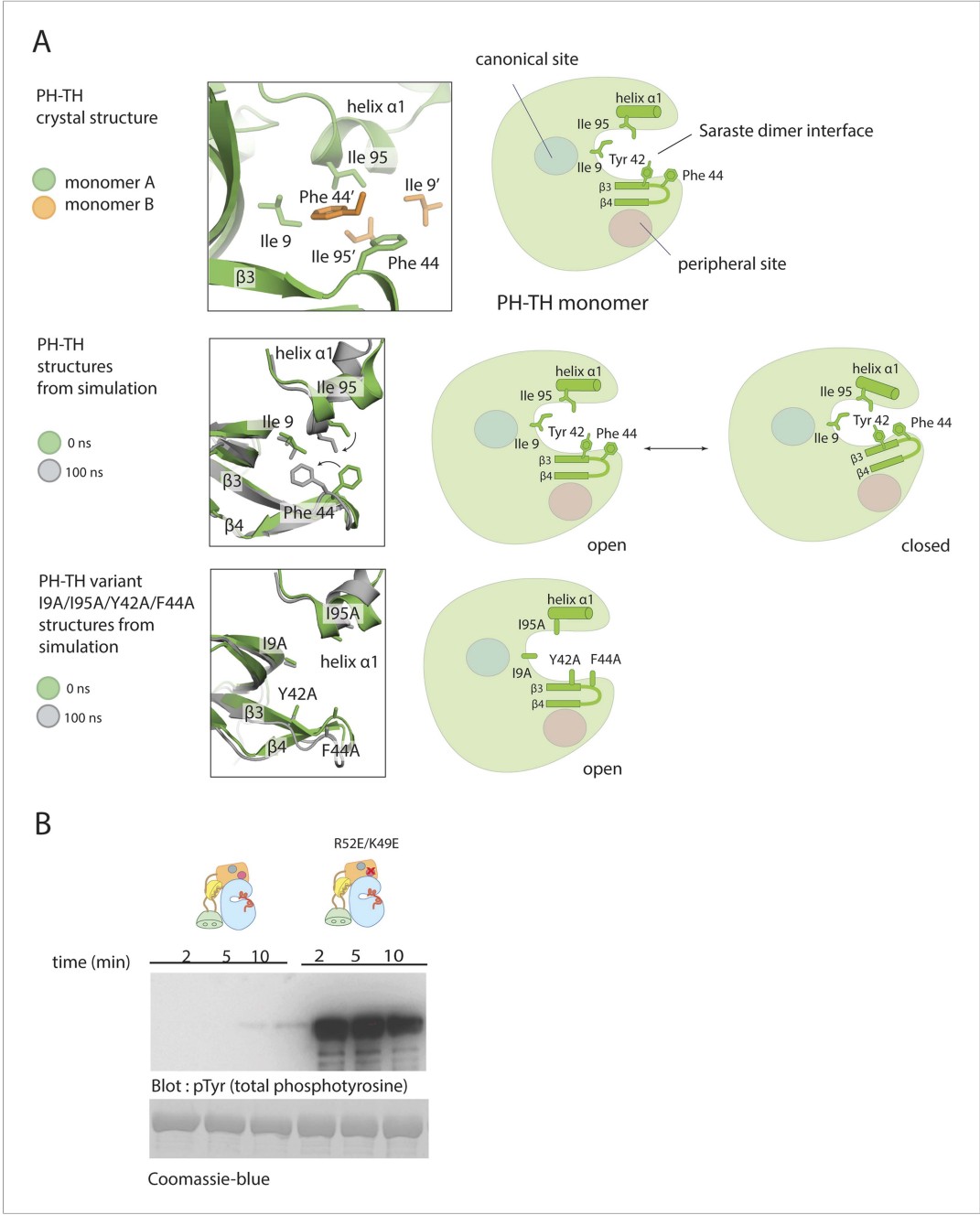

**Figure 9**. Possible allosteric and electrostatic effects of binding IP$_6$. (**A**) Fluctuations of the PH-TH module during a molecular dynamic simulation. An instantaneous structure (t = 100 ns) from a 100 ns simulation of the PH-TH module shows a conformational change that closes the Saraste dimer interface (middle panel). Replacing these hydrophobic residues (Ile 9, Tyr 42, Phe 44 and Ile 95) with alanine prevents the dimer interface from closing in the simulations (lower panel). Although the β3/β4 loop becomes more dynamic, it remains open (see **Figure 9–figure supplement 1**). Binding of IP$_6$ may shift the equilibrium between open and closed conformations. (**B**) Activation of the Btk R52E/K49E mutant (2 μM). Arg 52 and Lys 49 are two residues in the peripheral binding site. Mutation of these two residues to glutamate reduces the local charge at the peripheral site by a net four units, and results in Btk activation.

The following figure supplement is available for figure 9:

**Figure supplement 1**. Flexibility in the PH domain of Btk and simulation of the kinetics of Btk autophosphorylation.

## Conclusions

Our studies on Btk have revealed the structural basis for its autoinhibition. One key finding is that the Src-like module of Btk adopts a compact and autoinhibited form that resembles the autoinhibited structures of the Src kinases and c-Abl, in which the SH3 domain binds to the SH2-kinase linker and holds the kinase domain in an inactive conformation. Our structure of the PH-TH-kinase construct of Btk shows that the PH-TH module stablizes the inactive conformation of the kinase domain, as well as the assembled conformation of the Src-like module of Btk.

A second, and quite unexpected, finding is that $IP_6$ can activate Btk strongly by binding to a newly identified site on the PH domain, probably by promoting transient dimerization of the PH-TH module. Because $IP_6$ synthesis requires the precursor molecule $IP_3$, which is itself produced as a result of Btk activation, a transient elevation of $IP_6$ may create a positive feedback loop for Btk activation, either on the membrane or in the cytoplasm. Formation of the Saraste dimer interface is compatible with the simultaneous engagement of $PIP_3$ by both PH domains in the dimer, raising the possibility that $IP_6$ can stimulate dimerization of membrane-tethered Btk (*Figure 10*). $IP_6$ binding to the peripheral binding site may also enhance $PIP_3$ affinity in the canonical binding site and sensitize Btk to $PIP_3$ levels. The linker connecting the PH-TH unit to the SH3 domain promotes dimerization of Btk by interacting with the SH3 domain of a second molecule, providing an additional mechanism for Btk dimerization at high local concentration (*Laederach et al., 2002*). High levels of $IP_6$ may down-regulate the amount of Btk on membranes by competing for $PIP_3$ binding in the canonical binding site, as proposed for Akt1 regulation by $IP_7$ (*Chakraborty et al., 2010*).

The Src-like modules of Btk and Itk have 55% sequence identity, with all the residues critical for maintaining the assembled and autoinhibited Src-like conformation of Btk conserved in Itk (*Figure 10—figure supplement 1*). The Src-like module of Itk is therefore likely also to adopt the assembled Src-like conformation when inactive.

The roles of the PH-TH module in regulating Btk activity are not likely to be conserved in Itk. The four residues at the peripheral binding site that coordinate $IP_6$ in Btk are not conserved in Itk, unlike those that coordinate $PIP_3/IP_4$ at the canonical lipid-binding site. This lack of conservation also extends to the key residues at the inhibitory interface between the PH-TH module and the kinase domain in Btk and to residues at the Saraste dimer interface (*Figure 10—figure supplement 1*). A different soluble inositol phosphate, $IP_4$, promotes $PIP_3$ binding by the Itk PH-TH module; this activity is important for Itk regulation in T-cells (*Huang et al., 2007*). Thus, regulation by soluble inositol phosphates may be a general feature of Tec family kinases, but the specific effector and the molecular mechanism of the kinase response may be different in each case. The regulation of tyrosine kinases by soluble second messengers has not received much attention in the past. Our discovery that $IP_6$ activates Btk, along with the known effect of $IP_4$ on Itk, points to the importance of further studies on these and soluble effectors.

# Materials and methods

## Protein expression and purification

### Insect cell expression, purification, and characterization of mouse Btk

DNA for mouse Btk (residues 214 to 659; the Src-like module) was used to prepare baculovirus. Infected Sf21 cells were harvested by centrifugation 72 hr post-infection, then frozen in liquid nitrogen for storage. Cells were later thawed, resuspended in 25 mM Tris·Cl, pH 8.0, 50 mM NaCl, 5 mM DTT, 5 mM benzamidine, and complete protease inhibitor tablets (Roche Diagnostics, Switzerland), and lysed mechanically by Dounce homogenization followed by brief sonication. All remaining steps were carried out at 4°C. Cellular debris was cleared by centrifugation at 96,000×g for 1 hr, after which the supernatant was passed through a 0.45-μm filter. Cleared supernatant was loaded on to DEAE sepharose pre-equilibrated with 25 mM Tris·Cl, pH 8.0, 25 mM NaCl, and 5 mM DTT. Protein was eluted by a linear gradient to 25 mM Tris·Cl, pH 8.0, 0.35 M NaCl, 5 mM DTT over 3.7 column volumes, with the Src-like module of Btk eluting at approximately 2.6 column volumes.

Fractions containing the Src-like module of Btk were identified by Western blot, pooled, brought to 0.7 M ammonium sulfate, then loaded on to a phenyl sepharose column pre-equilibrated with 50 mM Tris·Cl, pH 8.0, 0.7 M ammonium sulfate, and 5 mM DTT. Contaminating protein was removed by an initial step to 0.2 M ammonium sulfate and the column eluted with a linear gradient to 50 mM Tris·Cl, pH 8.0 and 5 mM DTT over four column volumes. The Src-like module of Btk eluted as a broad peak starting at approximately 0.18 M ammonium sulfate.

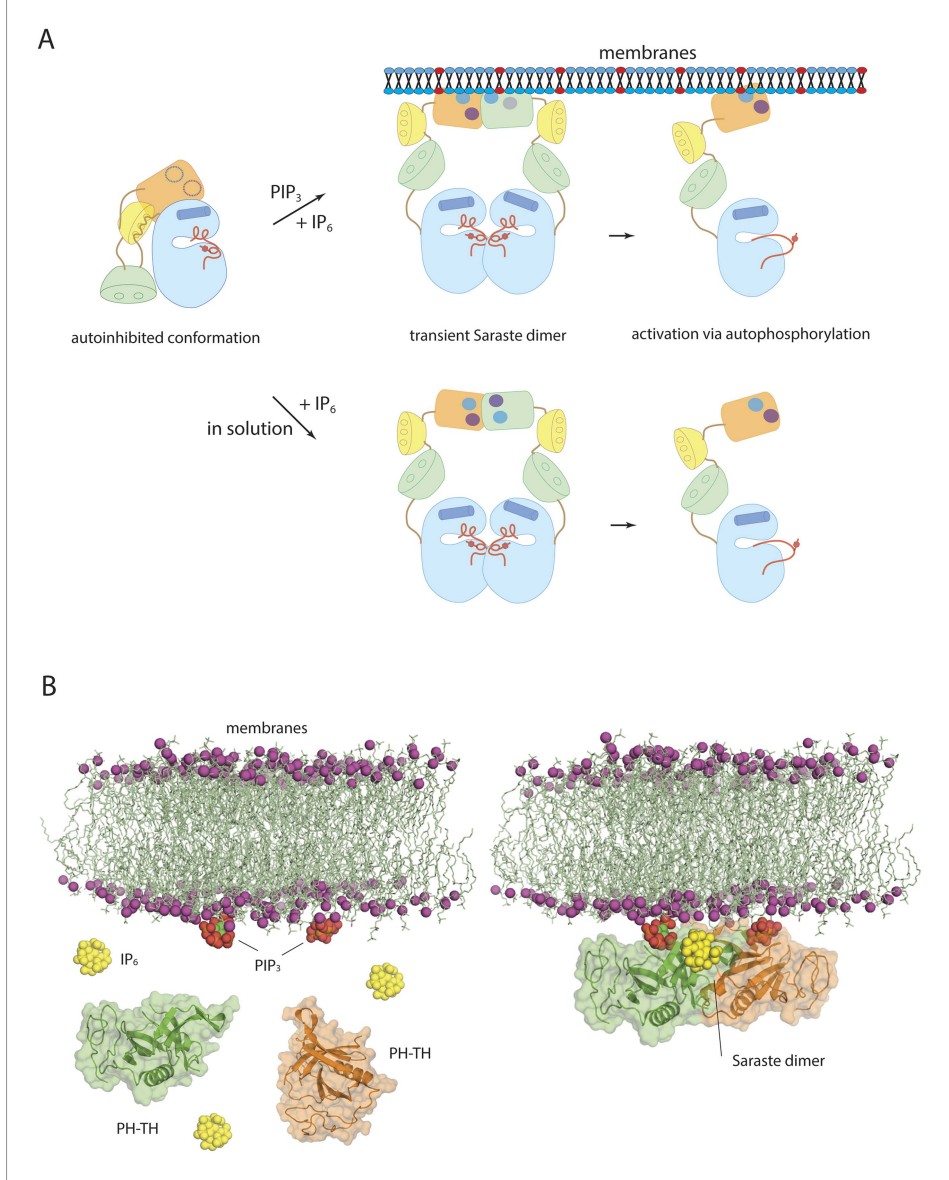

**Figure 10**. Models of Btk autoinhibition and activation. (**A**) The PH-TH module stabilizes the assembled conformation of the Src-like module of Btk in its inactive conformation. The activation of Btk involves formation of a transient dimer through the PH-TH module, which promotes trans-autophosphorylation and activates Btk. The stimulation of dimerization could occur at the membrane or in solution, as in our experiments. (**B**) $IP_6$ may stimulate membrane-tethered Btk. Formation of the Saraste dimer is compatible with the simultaneous engagement of $PIP_3$ by both PH domains in the dimer, and is compatible with the binding of $IP_6$ at the peripheral binding site. This raises the possibility that $IP_6$ can stimulate dimerization of the membrane-tethered Btk.

The following figure supplement is available for figure 10:

**Figure supplement 1**. Sequence alignment of Btk and Itk.

Fractions containing significant amounts of the Src-like module of Btk, identified by SDS PAGE, were pooled and concentrated using Centricon Plus-80 concentrators (Millipore, Billerica MA). Concentrated protein was dialyzed against 4 × 1 liter changes of 25 mM Tris·Cl, pH 8.5, 25 mM NaCl, 5 mM DTT. That buffer, excepting changes to the NaCl concentration, was maintained through the rest of the purification. After dialysis, protein was loaded on to a mono-Q HR 10/10 column, and

eluted with a linear gradient from 50 mM NaCl to 250 mM NaCl over twenty column volumes, with the protein eluting as two peaks at approximately 100 mM NaCl (peak A) and 150 mM NaCl (peak B), respectively. Fractions from the two peaks were pooled separately. Fractions from peak B were dialyzed against buffer containing 20 mM NaCl. The two pools could then be concentrated using Centricon-10 concentrators (Millipore). They were loaded separately onto a HiPrep S-200 16/60 size exclusion column pre-equilibrated in buffer containing 25 mM NaCl. Peak A eluted at the position corresponding to a globular protein of molecular mass 55 kDa, the value expected for a monomer of the Src-like module. Peak B eluted at a position corresponding of 140 kDa, compatible with an extended dimer. The protein in peak A could be concentrated using Centricon concentrators to approximately 20 mg/ml, while peak B required dialysis against a buffer containing no more than 10 mM NaCl, followed by concentration in dialysis tubing laid on PEG 20,000 at 4°C. In this way, the dimeric protein could be concentrated to approximately 10 mg/ml.

Peaks A and B were definitively identified as the Src-like module of Btk monomer and dimer, respectively, by chemical cross-linking using disuccinimidyl glutarate (Pierce Biotechnology, Waltham MA) and PAGE analysis (results not shown). All column media and pre-packed columns were purchased from Amersham Biosciences.

## Bacterial expression and purification of full-length bovine Btk and the PH-TH-kinase construct

DNA encoding bovine Btk (residues 1 to 659) was cloned into pET-28 expression plasmid yielding N-terminally 6-histidine-tagged SUMO-fusion protein. The hexahistidine-SUMO tag was cleavable by Ulp-1 SUMO-specific protease. For protein expression, the plasmid was transformed into BL21 DE3* (Novagen, Australia) cells that also contain the expression plasmids for GroES/GroEL and phosphatase YopH (Seeliger et al., 2005), and then grown in Terrific Broth (TB) media supplemented with 50 mg/l kanamycin, 50 mg/l streptomycin, and 50 mg/l chloramphenicol at 37°C. At a cell density corresponding to an absorbance of 1.5 at 600 nm, the temperature was reduced to 18°C, and protein production was induced by mixing the bacterial culture with equal volume of 4°C TB media supplemented with 1 mM IPTG, 100 μM ZnCl$_2$, 50 mg/l Kanamycin, 50 mg/l streptomycin, and 50 mg/l chloramphenicol. After 16 hr, cells were harvested by centrifugation, resuspended in Ni-NTA buffer A (500 mM NaCl, 20 mM imidazole, 25 mM Tris pH 8.5, 5% glycerol, and 1 mM DTT), and flash-frozen in liquid nitrogen. Cells were then thawed on ice and supplemented with a mixture of protease inhibitors (aprotinin 2 μg/ml, benzamidine 15 μg/ml, leupeptin 2 μg/ml, PMSF 1 mM, pepstatin A, 1 μg/ml).

After cell lysis by French press and removal of cell debris by centrifugation, clear lysates were loaded onto a HisTrap Ni-NTA column (GE Healthcare, UK), washed with 100 ml Ni-NTA buffer A. Proteins were eluted with a single step of Ni-NTA buffer A supplemented with 250 mM imidazole. Proteins were buffer exchanged into storage buffer (25 mM Tris-HCl, pH 8.0, 400 mM NaCl) by FastFlow desalting column (GE Healthcare), and affinity tags were removed by incubation with Ulp-1 protease at 4°C overnight. Cleaved proteins were collected in the flow-through during Ni-NTA affinity chromatography and were subjected to size-exclusion chromatography on a Superdex 200 column (GE Healthcare) equilibrated in storage buffer. Proteins were concentrated on an AmiconUltra centrifugal filter device (50 kDa cutoff; Millipore) to a final concentration of ~400 μM. Protein aliquots were frozen in liquid nitrogen and stored at −80°C.

The Src-like module (residues 217 to 659), SH2-kinase module (residues 270 to 659), and the kinase domain (residues 394 to 659) of bovine Btk, various bovine Btk PH-TH-kinase constructs and the PH-TH-Abl fusion construct (BTK residues 1 to 216 fused to human c-Abl residues 140 to 518) were all expressed and purified using the same protocol. The PH-TH module of bovine Btk (residues 1 to 172) was prepared in the same way as described above expect that the co-expression of GroEL/GroES and YopH is not necessary.

## Crystallization and structure determination

### The Src-like module of mouse Btk

The crystals of the dimeric protein (peak B) were grown using hanging-drop vapor diffusion at 4°C by mixing purified protein at 5 mg/ml in 25 mM Tris·Cl, pH 8.5, 10 mM NaCl, 5 mM DTT with an equal volume of the precipitant solution, 100 mM Tris·Cl, pH 8.5, 200 mM Na·acetate, 7.5% PEG 4000.

Wedge-shaped crystals grew to approximately 0.1 mm on a side after a period of 2 weeks. These crystals could be frozen by first transferring them, in four sequential steps, to a solution of 100 mM Tris·Cl, pH 8.5, 200 mM Na·acetate, 15% PEG 4000, followed by transfer in six sequential steps to a solution containing 100 mM Tris·Cl, pH 8.5, 200 mM Na·acetate, 15% PEG 4000, 30% glucose. The monomeric protein (peak A) did not yield crystals.

Data were collected at the Cornell High Energy Synchrotron Source (CHESS), beamline A1. Anomalous data, when present, were collected by the inverse beam method. Four sets of data contributed to the structure determination, collected from crystals in 5 mM $KAu(CN)_2$, 1 mM dimercuriacetate (DMA), 3 mM $Au(NO_3)_4$, and no adduct, respectively (*Table 2*). A heavy-atom partial structure was solved by inspection of $KAu(CN)_2$ anomalous difference Patterson maps, which revealed two heavy binding sites. Heavy-atom positions were then refined using SHARP (*Bricogne et al., 2003*), and difference electron density maps were used to locate a single heavy-atom position in the DMA derivative. Individual mercury atoms within the DMA molecule could not be resolved due to high B-values, so the heavy-atom compound was modeled as a point scatterer. Other derivatives were not useful for MIR phasing, due either to a lack of isomorphism or the presence of overlapping heavy-atom sites. Density modification was performed after heavy-atom refinement in SHARP using the default solvent flipping protocol (*Bricogne et al., 2003*).

The kinase domain was built into the density modified maps with reference to the already determined Btk kinase domain structure (PDB ID 1K2P), after separately placing the large and small lobes. The SH3 domain was built by comparison with the c-Src SH3 domain (PDB ID 2SRC).

We generated a homology model for the SH2 domain from the SH2 domains present in PDB entries for Src, Hck and Abl containing SH3-SH2-kinase constructs (PDB IDs: 1FMK, 1KSW, 2SRC, 2PTK, 1AD5, 1QCF, 2HCK, 1OPK). This model was placed by hand, aligning helices αA and αB into the unambiguous density, then refined as a rigid-body using a 4.0 Å resolution cutoff. Multiple crystal averaging across four distinct data sets ($KAu(CN)_2$, DMA, $Au(NO_3)_4$ and native, using the program DMMULTI (*Cowtan and Main, 1993*; *Winn et al., 2011*), revealed the domain swap in the secondary β-sheet portion of the SH2 domain.

The structure was refined using alternating cycles of manual rebuilding in the program O (*Jones et al., 1991*) and torsional simulated annealing and grouped B-factor refinement against data to 2.8 Å using the program CNS_SOLVE (*Brunger et al., 1998*; *Brunger, 2007*). A final refinement of the model was performed using PHENIX (*Adams et al., 2002*) to incorporate TLS as three groups, defined as residues 214–260, 261–391, and 392–657. These groups were identified by the automated TLS group search procedure and agreed well with the character of the experimental maps. In all refinement runs in which the SH2 domain was present, an initial temperature factor was refined for the entire SH2 domain, and then the SH2 domain was fixed. SH2 domain geometry is thus strictly a result of the geometry of the initial homology model. Data collection and refinement statistics are summarized in *Table 2*. Illustrations were made in Pymol (DeLano Scientific, New York NY).

## Structure of the PH-TH-kinase construct of bovine Btk

The bovine PH-TH-kinase construct (residues 1 to 170 connected to residues 384 to 659) was thawed on ice and mixed with twofold excess $IP_3$ (Avanti lipid) and inhibitor CGI1746 (Selleck Inc, Houston TX), and was incubated at 4°C for 30 min before setting up trays. $IP_3$ binds to the protein, but does not activate it. Although present in the crystallization mix, it is not visible in the final electron density (see below). Needle clusters of crystals were obtained by sitting drop vapor diffusion by mixing equal volume of protein solutions at 15 mg/ml with reservoir solution containing 200 mM NaCl, 50 mM $MgCl_2$, and 20% PEG3350. To prepare seeds for microseeding, needle clusters were harvested and combined from 20 drops and transferred into a 1.5-ml eppendorf tube.

The bovine PH-TH-kinase construct with activation loop mutations (*Joseph et al., 2013*) was prepared in the same way as was the wild-type protein. Equal volumes of protein solution and the reservoir solution containing 200 mM NaCl, 50 mM $MgCl_2$, and 20% PEG3350 were mixed and were equilibrated in a hanging drop tray for 24 hr at 4°C. Microseeding was then performed by striping the wild-type crystal seeds using a cat whisker into the equilibrated drops. Large rock-like crystal clusters appeared ~4 days after micro-seeding, and single crystals (typical dimensions of 0.20 mm × 0.15 mm × 0.25 mm) can be isolated. For cryo-protection, crystals were soaked in reservoir solution supplemented with 25% glycerol, and flash-frozen in liquid nitrogen and stored at 100 K.

Data were collected at the Lawrence Berkeley National Laboratory, Advanced Light Source (ALS), beamline 8.2.1. Data reduction was carried out with the software package HKL2000 (*Otwinowski and Minor, 1997*). The structure was determined by molecular replacement using PHENIX (*Adams et al., 2002*) with two search models. These were the structure of the Btk kinase domain that has mutations in the activation loop (see below) and the structure of the PH-TH module. Refinement in PHENIX and COOT (*Emsley and Cowtan, 2004*) yielded the final models. No density for IP$_3$ is seen in difference electron density maps, even at low contour levels (2σ above the mean in σ weighted |F$_o$| − |F$_c$| difference maps). Data collection and refinement statistics are summarized in *Table 1*. Illustrations were made in Pymol (DeLano Scientific).

## Structure of the Btk PH-TH module bound to IP$_6$

The bovine Btk PH-TH module (residues 1 to 172) sample was thawed at room temperature and was incubated with fivefold excess IP$_6$ (Sigma) for 10 min. Plate-like crystals were obtained by sitting drop vapor diffusion by mixing equal volume of protein sample solution and the reservoir solution containing 25% PEG1500 and 0.1 M DL-malic acid pH 5.5. Crystals were then transferred to cryo-protection solution, which is the reservoir solution supplemented with 30% glycerol and 3 mM IP$_6$.

Data were collected at the Lawrence Berkeley National Laboratory, Advanced Light Source (ALS), beamline 8.2.1. Data reduction was carried out with the software package HKL2000 (*Otwinowski and Minor, 1997*). The structure was determined by molecular replacement using PHENIX (*Adams et al., 2002*) with the structure of the PH-TH module as the search model (PDB: 1BTK) (*Hyvönen and Saraste, 1997*). Refinement in PHENIX and COOT (*Emsley and Cowtan, 2004*) yielded the final models. The structural restraint file for IP$_6$ was generated using REEL in PHENIX (*Adams et al., 2002*) based on the conformation of IP$_6$ seen in the structure of Auxin receptor TIR1 (PDB: 2P1P) (*Tan et al., 2007*). Data collection and refinement statistics are summarized in *Table 1*. Illustrations were made in Pymol (DeLano Scientific).

## Structure of the Btk kinase domain activation loop mutant

Btk kinase domain (residues 395 to 659) with the activation loop mutations (L542M, S543T, V555T, R562K, S564A, and P565S) (*Joseph et al., 2013*) was thawed at room temperature and was incubated with twofold excess CGI1746 (Selleck Inc) and fivefold excess IP$_6$ (Sigma) for 10 min. Crystals were obtained by sitting drop evaporation methods by mixing equal volume of protein sample solution and the reservoir solution containing 0.2 M ammonium citrate dibasic and 20% PEG 3350. Crystals were then transferred to cryo-protection solution which is the reservoir solution supplemented with 30% glycerol and 3 mM IP$_6$.

Data were collected at the Lawrence Berkeley National Laboratory, Advanced Light Source (ALS), beamline 8.2.1. Data reduction was carried out with the software package HKL2000 (*Otwinowski and Minor, 1997*). The structure was determined by molecular replacement using PHENIX (*Adams et al., 2002*) with the structure of wild-type Btk kinase domain (PDB: 3OCS) (*Di Paolo et al., 2011*). Refinement in PHENIX and COOT (*Emsley and Cowtan, 2004*) yielded the final models. No density can be located for IP$_6$ in crystals. Data collection and refinement statistics are summarized in *Table 1*.

### Preparation of autophosphorylated Btk

Btk samples were diluted to 4 µM using the buffer that contains 150 mM NaCl, 25 mM Tris, pH 7.5, and 5% glycerol. For experiments involving lipid vesicles and inositol phosphates, additional 500 µM lipid and 200 µM inositol phosphates are supplemented into the dilution buffer, respectively, followed by a 10-min incubation at room temperature.

The diluted samples were mixed with equal volume of the reaction buffer that contains 150 mM NaCl, 20 mM MgCl$_2$, 2 mM ATP, 25 mM Tris, pH 7.5, 2 mM sodium vandate, and 5% glycerol and incubated at room temperature. The final concentration of Btk samples is 2 µM. At each time point, the samples were either frozen in liquid nitrogen and stored at −80°C for the coupled kinase assays, or mixed with equal volume of quench buffer that contains 2X SDS-PAGE buffer supplemented with 100 mM EGTA, followed by incubating at 95°C in a heat block for 15 min for western blot assays.

### Coupled kinase assay

A continuous pyruvate-kinase coupled assay was performed to measure the kinase activity of the proteins as described (*Barker et al., 1995*), with minor modifications. The ATP concentration

was kept at 1 mM in all the assays. The buffer used contains 150 mM NaCl, 10 mM $MgCl_2$, 25 mM Tris, pH 7.5, 2 mM sodium vandate. The protein concentrations were kept at 1 µM for all Btk constructs. The peptide sequence is ERDINSLYDVSRMYVDPSEIN, which was derived from residues 746 to 766 of PLC-γ2. The $K_m$ value for phosphorylated full-length Btk for this peptide is $925 \pm 117$ µM, and the peptide concentration was kept at 1 mM in all experiments. The reactions were performed in a 96-well plate and were monitored by SpectraMax (Molecular Devices, Sunnyvale CA). The ATP turn-over rate ($v_0$: $min^{-1}$) was calculated by fitting the data acquired from first 150 s using a linear regression model.

## Western blot

The levels of autophosphorylation of purified Btk samples were monitored using non-specific anti-phosphortyrosine antibody 4G10 (EMDMillipore). The total amount of Btk was monitored using coomassie-blue stained SDS-PAGE. For each set of experiments, 10 µl of each Btk sample was loaded on two 12% SDS-PAGE gels. The two gels were run at 250 V, 400 mA for 35 min at room temperature.

One gel was transferred to 50 ml fixing buffer containing 50% ethanol and 10% acetic acid and was heated for 30 s in a microwave. The gel is then transferred to 50 ml staining buffer that contains 5% ethanol, 7.5% acetic acid, and 0.00025% coomassie brilliant blue R-250 and was heated for another 30 s in a microwave. The gel was then left to cool at room temperature in the staining buffer on a rocker for 1 hr and was then imaged using ImageLab (Bio-Rad, Hercules CA).

The other gel was transferred into western blot transfer buffer that contains 25 mM Tris at pH 7.4, 192 mM glycine, and 20% methanol and was incubated at room temperature for 15 min. Protein samples were then transferred to PVDF membranes from the gel using the semi-dry transfer device TRANS-BLOT (Bio-Rad). The membranes were then blocked for 1 hr in TBST buffer (20 mM Tris pH 7.5, 150 mM NaCl, 0.1% Tween-20) supplemented with 5% dry milk, and incubated with primary antibody 4G10 (1:2000 dilution) at 4°C overnight. The membranes were then washed three times in TBST buffer, and incubated with HRP-linked anti-mouse IgG antibody (1:5000 dilution) in TBST buffer at room temperature for 1 hr. The membranes were then washed three times in TBST buffer, incubated with 0.5 ml WesternBright Quantum (Advansta, Menlo Park CA) for 1 min, and were imaged using ImageLab (Bio-Rad). The western blots were reproduced at least three times and the most representative ones were chose to present.

## Small unilamellar vesicle preparation

Three lipids, 18:1 DOPS, 18:1 (Δ9-Cis)(DOPC), and 18:1 PI(3,4,5) $P_3$ (Avanti, Alabaster AL) were dissolved in chloroform and stored at −20°C. For liposome preparations, the three lipids are mixed in a test tube at the desired molar ratio (10:75:5); chloroform was evaporated under a nitrogen atmosphere for 15 min; the samples were transferred to a vacuum desiccator and dried overnight at room temperature. Dry films were hydrated in buffer containing 25 mM Tris-HCl (pH 7.4) and 100 mM NaCl to a concentration of 10 mg/ml.

The hydrated liposomes are heterogeneous in size and the diameters of the liposomes vary from ~50 nm to ≥ 1 µm. To prepare small unilamellar vesicles, hydrated liposomes were subject to seven freeze–thaw cycles using liquid nitrogen, followed by extrusion through 100-nm filters (Avestin, Canada). The liposomes prepared using this method have diameters around 100 nm, as measured using negative staining electron microscopy (data not shown).

## Isothermal titration calorimetry

The isothermal titration calorimetry experiments were performed using MicroCal-autoITC 200 (GE Healthcare). Various Btk samples are diluted in the binding buffer containing 150 mM NaCl, 25 mM Tris, pH 7.5, and 5% glycerol to a final concentration of 20 µM. $IP_4$ and $IP_6$ were diluted using the same buffer to a final concentration of 300 µM. Each set of ITC experiments included three samples: 420 µl of Btk protein sample in the cell, 150 µl of $IP_4$ or $IP_6$ at a concentration 300 µM in the syringe, and 500 µl of binding buffer. All samples are stored at 4°C before the titration experiments.

The ITC experiments were performed at 20°C. An initial injection of 0.5 µl was excluded from data analysis, followed by 14 injections of 3 µl each, separated by 180 s with a filter period of 5 s. The protein solution was stirred at 500 rpm over the course of titration. Titration curves were fit with a one-site binding model. The three fitting parameters are stoichiometry, N, association constant, $K_a$ and

binding enthalpy, $\Delta H$. The binding entropy is then calculated using formula: $\Delta S = (\Delta H + RT\ln K_a)/T$. Thermodynamic parameters for various Btk constructs binding to IP$_4$ and IP$_6$ are listed in *Table 1*.

## Molecular dynamics simulations

Molecular dynamics trajectories were generated using the Gromacs 4.6.2 package (*Pronk et al., 2013*) using the ff99SB-ILDN force field (*Lindorff-Larsen et al., 2010*). All simulations were in water, using the TIP3P water model; appropriate counterions (Na$^+$ and Cl$^-$) were added to neutralize the net charges. After initial energy minimization, the systems were subjected to 100 ps of constant number, volume and temperature (NVT) equilibration, during which the system was heated to 300 K. This was followed by a short equilibration at constant number, pressure, and temperature (NPT, 100 ps). Finally, the production simulations were performed under NPT conditions, with v-rescale thermostats in Gromacs 4.6.2, respectively, in the absence of positional restraints. Periodic boundary conditions were imposed, and particle-mesh Ewald summations were used for long-range electrostatics and the van der Waals cut-off is set at 10 Å. A time step of 2 fs was employed and the structures were stored every 2 ps. His 143, Cys 154, Cys 155, and Cys 165, which coordinate Zn$^{2+}$ in the PH domain were set to be deprotonated. No additional positional restrains are applied to the Zn$^{2+}$ ion, and its sp$^3$ tetrahedral coordination remains intact during all MD simulations.

The visual molecular dynamics (VMD) analysis toolkit (*Humphrey et al., 1996*) was used to make some of the root mean square deviation (RMSD) measurements, others were done using PyMol.

## Homology modeling

The active conformation of the Btk kinase domain was modeled based on that of the Lck kinase domain (PDB: 3LCK) (*Sicheri et al., 1997*), using Modeller (*Eswar et al., 2006*). The sequence of the kinase domains of Lck and Btk were aligned by ClustalW (*Larkin et al., 2007*). We chose the model with the lowest Discrete Optimized Protein Energy (DOPE: −0.84) as representative of the active conformation of Btk.

## Estimate of membrane-bound Btk concentration

We estimated the effective concentration ($C$) of Btk on a lipid vesicle by the following equation:

$$C = \frac{N}{V \times N_a}.$$

$N$ is the number of Btk molecules on a lipid vesicle, $N_a$ is the Avogadro number, and $V$ is the effective volume in which Btk moves on a lipid $N_0 = \frac{A}{A_0}$ vesicle. An estimate for $V$ is given by:

$$V = A \times L.$$

Here, $A$ is the surface area of a lipid vesicle and $L$ is the length of a Btk molecule, $\sim 100$ Å, as measured from the composite model of full-length Btk. Assuming all the PIP$_3$ molecules (molar fraction, $f$) are bound to Btk, then,

$$N = f \times N_0,$$

where $N_0$ is the total number of lipids on a vesicle. This is given by,

$$N_0 = \frac{A}{A_0},$$

in which $A_0$ is the surface area of a lipid molecule. We assume all lipids have the same surface area for their headgroups, which is roughly 0.6 nm$^2$ (*Gulik-Krzywicki et al., 1967*). Therefore $C$ is given by:

$$C = \frac{N}{V \times N_0} = \frac{f \times \frac{A}{A_0}}{A \times L \times N_a} = \frac{f}{A_0 \times L \times N_a}.$$

In our experiments, $f = 0.05$, A$_0$ = $\sim 60$ Å$^2$, N$_a$ = $6.03 \times 10^{23}$ mol$^{-1}$, and $L$ = $\sim 100$ Å, giving a value of 12 mM for C.

## Acknowledgements

We thank Arthur Weiss, Terri Kadlecek, William Degrado, and Manasi Bhate at UCSF for helpful discussions and help with ongoing experiments; Kuriyan lab members, especially Jon Winger, Jeff Iwig, and Qingrong Yan, for helpful discussions; Ross Wilson (UC Berkeley) for training on micro-ITC 200, David King (HHMI) for synthesizing peptides; Tony Iavarone (QB3) for mass spectrometry analysis; Tiago Barros and the beamline staff at ALS 8.2.1/8.2.2 for advice on data collection and structure refinement; Ming Lei, Wenqing Xu, and Michael Eck for contributions to the SH3-SH2-kinase crystallization and structure determination; Tiago Barros, Neel Shah, Meg Stratton, and Yongjian Huang for critically reading the manuscript. This work was supported in part by NIH grant PO1 AI091580 to JK. QW is supported by the Cancer Research Institute-Irvington Institute Fellowship Program.

## Additional information

### Competing interests

SCH: Reviewing editor, *eLife*. JK: Senior editor, *eLife*. The other authors declare that no competing interests exist.

### Funding

| Funder | Grant reference | Author |
| --- | --- | --- |
| Cancer Research Institute (CRI) | postdoctoral fellowship | Qi Wang |
| National Institutes of Health (NIH) | PO1 AI091580 | John Kuriyan |

The funders had no role in study design, data collection and interpretation, or the decision to submit the work for publication.

### Author contributions

QW, EMV, Conception and design, Acquisition of data, Analysis and interpretation of data, Drafting or revising the article; LMN, CER, JAZ, Acquisition of data, Analysis and interpretation of data; SCH, JK, Conception and design, Analysis and interpretation of data, Drafting or revising the article

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
