## [Decision Letter]

Thank you for sending your work entitled “Autoinhibition of Bruton's tyrosine kinase (Btk) and activation by soluble inositol hexakisphosphate” for consideration at *eLife*. Your article has been favorably evaluated by Tony Hunter (Senior editor) and 3 reviewers, one of whom is a member of our Board of Reviewing Editors.

The Reviewing editor and two other reviewers discussed their comments before we reached this decision, and the Reviewing editor has assembled the following comments to help you prepare a revised submission.

The manuscript by Qi Wang et al concerns a comprehensive biochemical analysis of the regulation of Btk (Bruton's tyrosine kinase) and its unexpected activation by IP_6_. Btk is of broad interest in immune signaling and as a drug target. This paper enlightens the community about several very interesting and important features regarding Btk regulation: 1) The kinase domain is held in an inhibited state by the SH2 and SH3 domains in a manner analogous to Src (and Abl);

2) Unique to Btk, its PH domain participates directly through the kinase N-lobe with the SH2 and SH3 domain to suppress the kinase domain;

3) Vesicle bound PIP_3_ can activate Btk by inducing dimerization;

4) Soluble IP_6_ can activate Btk by binding a non-canonical surface of the PH domain and promoting dimerization.

The comprehensive model of Btk regulation described in this manuscript derives from several high-resolution crystal structures, enzymologic analysis of autophosphorylation as well as peptide phosphorylation in the presence of various activators, site-directed mutagenesis, and molecular dynamics simulations. The discovery of IP_6_ regulation is perhaps the most special aspect of the manuscript and should generate substantial interest in the tyrosine kinase and pleckstrin homology domain fields. While the authors rightly cite Huang et al, Science (2007),which reports that soluble IP_4_ can activate ITK tyrosine kinase, our read of this study is that it minimally characterizes this putative activation, which in vitro was quite marginal. In summary, the current manuscript is an excellent fit for *eLife*. We ask the authors to consider the comments below.

There is no mention in the Introduction of the prior Btk structures reported. A search of the PDB reveals that there are over 30 structures of various Btk domains in active and inactive states. It would be helpful to the reader to briefly summarize the known Btk domain structures and include a statement indicating that although the structures of the individual Btk domains are established, the relationships between the kinase domain and the SH2, SH3 and PH-TH domains were not known prior to the current manuscript.

Major comments:

The role of IP_6_ as a glue holding neighboring PH domains together is somewhat reminiscent of a related role for soluble IP_4_/6 in stabilizing HDAC/corepressor complexes (CJ Millard et al, Mol Cell 51, 57-67,2013). The authors may wish to compare the Btk and corepressor inositol phosphate-containing structures and comment on the relationship if appropriate.

We were unable to find the specific peptide sequence of the kinase substrate or its concentration in the Methods. While not absolutely essential, it would also be informative to know what the Km of the peptide is and whether this changes in different states of Btk activation. If there were Km differences among particular Btk conformational states, for example IP_6_ rather than PIP_3_ activation, this could give rise to different cellular signaling outputs.

In their full-length Btk model, the SH3 domain and the PH domain appear to make electrostaic interactions with each other (Figure 4). Do the authors predict that this surface is modulated by IP_6_ or PIP_3_ binding? Likewise, is there any reason to think that ligation of the SH3 domain by external ligands impacts this SH3/PH interaction?

Although the authors show that PIP_3_-derived IP_4_ binds more tightly to the Btk PH domain than IP_6_ (Kd of 30 nM vs. 250 nM), it is interesting to consider that high concentrations (>20 µM) of soluble IP_6_ could conceivably compete Btk off of the PIP_3_ containing membrane. Thus, one way to consider the importance of the Btk allosteric binding site for IP_6_ is that it is needed to prevent down-regulation by competition of IP_6_ for PIP_3_ at the PIP_3_ binding site. If true, Btk membrane binding may be an overstated facet of its regulation, as the authors also suggest. It will be fascinating in future studies to investigate selective mutants in vivo to dissect the relative contributions of PIP_3_ and IP_6_ in signaling.

Alternatively, just as IP_6_ might stimulate dimerization of membrane-tethered Btk, as suggested in the Conclusions, might it not also enhance the affinity (avidity) of Btk for PIP_3_-containing membranes by promoting dimerization? This would have the effect of 'sensitizing' Btk to PIP_3_ levels, and could be important. Such an effect of IP_6_ (or IP_7_ or IP_8_) could also/alternatively counteract inhibition of PH domain function by competition at the canonical site (as suggested by Chakraborty et al for Akt: Cell 143, 897). It might thus selectively 'spare' Btk from such inhibition?

In the beginning of the Results and Discussion section, the authors describe the difference in the angles for the kinase SH3 domain with respect to other kinases and they explain the differences in orientation of the SH2 and SH3 domains, saying: “Coupling between the SH3 and SH2 domains may therefore be somewhat different in these kinases.” Although this may be true, it is also possible that there is not a fixed orientation in solution between the different domains and the differences reflect different arrangements captured by the crystals. The authors also say that: ”diffuse electron density for the SH2 shows that this part of the polypeptide chain occupies a range of positions and adopts multiple conformations“. This statement seems to support such an idea.

Also in the Results and Discussion section, the authors say that: “… nearly identical to those determined previously using NMR”. They should include RMSD for the C-alphas that were aligned.

This sentence, in the same section of the text: “The domain swap occurs in the same location within the SH2 domain as a similar swap seen in a Grb2 SH2-domain crystal structure.” From the drawing this similarity is not so apparent. They use the same helix against the same beta sheet, but the orientation of the two monomers appears different. In Grb2 the helices are essentially parallel to the beta-sheet whereas in Btk, these helices seem to be perpendicular.

The authors say that it is unlikely “that a domain-swapped dimer” is physically important. That being the case, the authors may wish to consider reducing the overall discussion about domain swapping.

In the Results and Discussion, it says that “r.m.s. deviation from c-Src is 2.3 Å for the 24 α-carbons”. This seems to be a rather large deviation for so few carbons.

In that same section of the article, the authors call an Arg-Phe interaction an “amino-aromatic interaction”. Is this actually amino (cation)-aromatic pair, or is it a π-type (Pi-type) interaction between the guadinium of the Arg and the aromatic ring of the Phe?

In the subsection “Structure of a PH-TH-kinase construct of Btk”, it says: “The PH-TH module contacts the kinase N-lobe. The interface differs from the one found in Akt1”. Is this because they are intrinsically different or because of the deletion of the SH2 and SH3 domains? Some comment on this could be helpful.

Subsection headed “A structural model for full-length Btk”: it is not very clear what the cause is, but the deviations between the crystallographic structures and the ones at the end of the trajectories appear to be large. It is not very clear what these trajectories were probing and what the implications are for their having departed so significantly from the original structures?

The manuscript says, in the “Activation of Btk by IP_6_ in solution” section: “but these required higher protein concentration and longer times for appreciable autophosphylation than did IP_6_IP6.” Looking at Figure 5 the Coomassie-blue for some of the IP_5_ doesn't show higher protein concentration.

Subsection headed “The canonical lipid-binding pocket of the PH-TH module is not critical for IP_6_-mediated Btk activation”: when the authors note that R28C/N24E abolishes IP_4_ binding and weakens IP_6_ binding by 20-40-fold using ITC, they appear to conclude that the residual binding is solely due to the separate IP_6_ site. Is it possible that the double mutant still allows for IP_6_ binding to the IP_4_ site? Perhaps the authors can clarify this section a bit more.

Subsection headed “Structure of the IP_6_-bound PH-TH module”: it is not said, although one can assume, that the dimers are two-fold symmetric. If that is the case it is not very clear whether or not the open-ended chain is a consequence of the crystal lattice rather than being biologically relevant. Therefore, the binding site that is formed by two molecules in the context of the open-ended chain of dimers may not be present in solution. The authors appear to imply that the IP_6_ interaction site is mostly determined by one of the chains. If that were the case and IP_6_ binds to that site even in the absence of the dimer-formed open-ended chain, then shouldn't one have observed a more complex stoichiometry in the ITC experiments (which shows an n∼1)? More explanation here could be helpful.

The section on IP_6_ regulation is fairly lengthy given the degree of speculation. The authors could consider condensing this.

In the beginning of the Results and Discussion, it could be helpful to include a short introduction indicating which structures were determined as part of this work, to what resolution, and a reference to Table 2.

It would be interesting to know whether the orientation of the IP_6_ in the peripheral site is well defined or not. IP_6_ binding to such a 'flat' (as the authors put it) site is often heterogeneous in pose, but the data in Figure 5 suggest that this might not be the case here, as does the second paragraph in “The peripheral binding site is critical for IP_6_-mediated Btk activation”. Some comment on stereospecificity would be helpful for those who think about IP_6_ binding sites. Perhaps a density panel could be added to Figure 7 or its supplement?

In reflecting on the organization of the paper, one suggestion would be to place Figure 2 (“Structure of a PH-TH-kinase construct of Btk”) before Figure 1 (“Crystal structure of the Btk SH3-SH2-kinase module”). The story might flow/unfold better if the discovery of the potential autoinhibitory role of the PH-TH region leads into discovery of the Src-like disposition of domains in the Src-like module, then wrapped up nicely by the work in Figure 3 on whole Btk.

---

## [Author Response]

*There is no mention in the Introduction of the prior Btk structures reported. A search of the PDB reveals that there are over 30 structures of various Btk domains in active and inactive states. It would be helpful to the reader to briefly summarize the known Btk domain structures and include a statement indicating that although the structures of the individual Btk domains are established, the relationships between the kinase domain and the SH2, SH3 and PH-TH domains were not known prior to the current manuscript*.

We appreciated this point. We have added two sentences in the Introduction summarizing the previously determined structures of Btk fragments and have prepared a supplemental table (Table 1) that lists all the structures of Btk and other Tec family kinases in the Protein Data Bank, along with the accession codes.

Major comments:

*The role of IP*_*6*_
*as a glue holding neighboring PH domains together is somewhat reminiscent of a related role for soluble IP*_*4/6*_
*in stabilizing HDAC/corepressor complexes (CJ Millard et al, Mol Cell 51, 57-67, 2013). The authors may wish to compare the Btk and corepressor inositol phosphate-containing structures and comment on the relationship if appropriate*.

We have added a paragraph in the Results section discussing the role of IP_4_ in regulating the HDAC/corepressor complex, and compared that with the role of IP_6_ in regulating Btk activation. The mechanism of IP6-induced transient Btk dimerization is fundamentally different from that of the regulation of HDACs by IP_4_. We point out in the revised manuscript that in case of Btk, the peripheral IP_6_ binding site is located next to the Saraste dimer interface, and IP6 is not directly involved in any interactions between the two Btk subunits in the Saraste dimer.

*We were unable to find the specific peptide sequence of the kinase substrate or its concentration in the Methods. While not absolutely essential, it would also be informative to know what the Km of the peptide is and whether this changes in different states of Btk activation. If there were Km differences among particular Btk conformational states, for example IP*_*6*_
*rather than PIP*_*3*_
*activation, this could give rise to different cellular signaling outputs*.

We apologize for not including this information in the original manuscript. We have added the peptide sequence and the Km value for the peptide for phosphorylated full-length Btk in the Materials and methods section. We did not measure the Km values for various Btk states, and we agree that this could be an interesting subject for the future studies.

*In their full-length Btk model, the SH3 domain and the PH domain appear to make electrostaic interactions with each other (*Figure 4*). Do the authors predict that this surface is modulated by IP*_*6*_
*or PIP*_*3*_
*binding? Likewise, is there any reason to think that ligation of the SH3 domain by external ligands impacts this SH3/PH interaction?*

This point made us realize that we had not provided structural diagrams that clearly showed the location of the peripheral IP_6_ site with respect to other regulatory elements. We don’t think that IP_6_ or PIP_3_ binding will directly disassemble the inactive conformation of Btk, because both binding sites are located distal to the interface between the PH-TH module and the Src-like module. We have clarified this by marking both binding sites in our cartoon illustrations in all figures and have added corresponding sentences in the Results section. On the other hand, the ligation of the SH3 domain by external ligands will likely disassemble inactive conformation of Btk. We now note this in the Results section.

*Although the authors show that PIP*_*3*_*-derived IP*_*4*_
*binds more tightly to the Btk PH domain than IP*_*6*_
*(Kd of 30 nM vs. 250 nM), it is interesting to consider that high concentrations (>20 µM) of soluble IP*_*6*_
*could conceivably compete Btk off of the PIP*_*3*_
*containing membrane. Thus, one way to consider the importance of the Btk allosteric binding site for IP*_*6*_
*is that it is needed to prevent down-regulation by competition of IP*_*6*_
*for PIP*_*3*_
*at the PIP*_*3*_
*binding site. If true, Btk membrane binding may be an overstated facet of its regulation, as the authors also suggest. It will be fascinating in future studies to investigate selective mutants* in vivo *to dissect the relative contributions of PIP*_*3*_
*and IP*_*6*_
*in signaling*.

This is an excellent point. We agree with reviewers’ comments and have expanded our discussion on the potential roles of IP_6_ in regulating membrane-tethered Btk.

*Alternatively, just as IP*_*6*_
*might stimulate dimerization of membrane-tethered Btk, as suggested in the Conclusions, might it not also enhance the affinity (avidity) of Btk for PIP*_*3*_*-containing membranes by promoting dimerization? This would have the effect of 'sensitizing' Btk to PIP*_*3*_
*levels, and could be important. Such an effect of IP*_*6*_
*(or IP*_*7*_
*or IP*_*8*_*) could also/alternatively counteract inhibition of PH domain function by competition at the canonical site (as suggested by Chakraborty et al for Akt: Cell 143, 897). It might thus selectively 'spare' Btk from such inhibition?*

We appreciate this point as well, and have included this in the concluding section.

*In the beginning of the Results and Discussion section, the authors describe the difference in the angles for the kinase SH3 domain with respect to other kinases and they explain the differences in orientation of the SH2 and SH3 domains, saying: “Coupling between the SH3 and SH2 domains may therefore be somewhat different in these kinases.” Although this may be true, it is also possible that there is not a fixed orientation in solution between the different domains and the differences reflect different arrangements captured by the crystals. The authors also say that: “diffuse electron density for the SH2 shows that this part of the polypeptide chain occupies a range of positions and adopts multiple conformations”. This statement seems to support such an idea*.

We agree with this comment.

*Still in the Results and Discussion section, the authors say that: “… nearly identical to those determined previously using NMR”. They should include RMSD for the C-alphas that were aligned*.

We thank the reviewers for pointing this out. We have calculated the RMSD value as suggested by the reviewers and added it to the Results section.

*This sentence, in the same section of the text: “The domain swap occurs in the same location within the SH2 domain as a similar swap seen in a Grb2 SH2-domain crystal structure.” From the drawing this similarity is not so apparent. They use the same helix against the same beta sheet, but the orientation of the two monomers appears different. In Grb2 the helices are essentially parallel to the beta-sheet whereas in Btk, these helices seem to be perpendicular*.

The relative orientation of the two subunits in the Btk SH2 dimer is rotated by about 90° with respect to that of the Grb2 SH2 dimer, as noted by the reviewers. We have deleted the word “similar” and replaced it by “domain”. The sentence now reads: “The domain swap occurs in the same location within the SH2 domain as a domain swap seen in a Grb2 SH2-domain crystal structure.”

*The authors say that it is unlikely “that a domain-swapped dimer” is physically important. That being the case, the authors may wish to consider reducing the overall discussion about domain swapping*.

We have considered this point. We have, however, decided to retain the discussion because the domain swap was an unexpected feature of the crystal structure.

*In the Results and Discussion, it says that “r.m.s. deviation from c-Src is 2.3 Å for the 24 α-carbons”. This seems to be a rather large deviation for so few carbons*.

The activation loop is a highly flexible part of the kinase domain. For example, when the active and inactive conformation of these 24 Cα atoms are compared between an active (PDB: 3LCK) and inactive (PDB: 1QCF) Src kinase conformation, the rms deviation is 11 Å. Thus 2.3 Å indicates that the conformations of the activation loops are similar.

In that same section of the article, the authors call an Arg-Phe interaction an “amino-aromatic interaction”. Is this actually amino (cation)-aromatic pair, or is it a π-type (Pi-type) interaction between the guadinium of the Arg and the aromatic ring of the Phe?

We have confirmed that “amino-aromatic” is a better description for these interactions in the crystal structure.

*In the subsection “Structure of a PH-TH-kinase construct of Btk”, it says: “The PH-TH module contacts the kinase N-lobe. The interface differs from the one found in Akt1”. Is this because they are intrinsically different or because of the deletion of the SH2 and SH3 domains? Some comment on this could be helpful*.

We thank the reviewers for pointing out this potential confusing statement in our manuscript. The PHkinase interface in Btk is very different from that in Akt1, and we have clarified this by adding a couple of sentences in the corresponding sections in the Results.

*Subsection headed “A structural model for full-length Btk”: it is not very clear what the cause is, but the deviations between the crystallographic structures and the ones at the end of the trajectories appear to be large*. *It is not very clear what these trajectories were probing and what the implications are for their having departed so significantly from the original structures?*

We thank the reviewers for pointing this out and we apologize that we did not make this clear in the manuscript. The relatively large deviation arises from the pivoting motions of the molecule during molecule dynamic simulations, and the use of only the C lobe of the kinase to align these structures, which amplified the deviations. The overall conformation of Btk at the end of the trajectories is actually very similar to the initial structure. We have now explained this in the Result section, and state that the rms deviation between the initial and final structures is only 2.3 Å when the alignment is done on the entire Src-like module.

*The manuscript says, in the “Activation of Btk by IP*_*6*_
*in solution” section: “but these required higher protein concentration and longer times for appreciable autophosphylation than did IP*_*6*_*.” Looking at*
Figure 5
*the Coomassie-blue for some of the IP*_*5*_
*doesn't show higher protein concentration*.

Perhaps the confusion here arises because the results of two different experiments (two different gels) are shown in Panel 5B. The top gel in this panel has lower protein concentration (2 μM Btk), shorter reaction time (5 minutes) and shorter exposure time. The bottom gel has higher Btk concentration (4 μM Btk), longer reaction time (10 minutes) and the gel has been exposed for a longer time. The figure legend has been modified to make this clearer.

*Subsection headed “The canonical lipid-binding pocket of the PH-TH module is not critical for IP*_*6*_*-mediated Btk activation”: when the authors note that R28C/N24E abolishes IP*_*4*_
*binding and weakens IP*_*6*_
*binding by 20-40-fold using ITC, they appear to conclude that the residual binding is solely due to the separate IP*_*6*_
*site. Is it possible that the double mutant still allows for IP*_*6*_
*binding to the IP*_*4*_
*site? Perhaps the authors can clarify this section a bit more*.

In ITC experiments, we saw no detectable binding when we added IP6 to a PH-TH module with mutations at both the canonical and the peripheral sites, which confirms that the residual binding of IP_6_ to the IP_4_-binding deficient mutant is due to binding at the peripheral binding site. This is stated in the subsection “The peripheral binding site is critical for IP_6_-mediated Btk activation”.

*Subsection headed “Structure of the IP*_*6*_*-bound PH-TH module”: it is not said, although one can assume, that the dimers are two-fold symmetric. If that is the case it is not very clear whether or not the open-ended chain is a consequence of the crystal lattice rather than being biologically relevant. Therefore, the binding site that is formed by two molecules in the context of the open-ended chain of dimers may not be present in solution. The authors appear to imply that the IP*_*6*_
*interaction site is mostly determined by one of the chains. If that were the case and IP*_*6*_
*binds to that site even in the absence of the dimer-formed open-ended chain, then shouldn't one have observed a more complex stoichiometry in the ITC experiments (which shows an n∼1)? More explanation here could be helpful*.

We thank the reviewers for pointing out this potentially confusing point, which was not well addressed in the original manuscript. We agree that this chain of dimers is unlikely to be biologically relevant, and the dimer that matters for the Btk activation is the Saraste dimer, which we discuss extensively in the paper. We have now clarified this by reducing the discussion of the open-ended chain of dimers in the Results section, and adding several sentences to make it clear that the open-ended chain is likely to be only a feature of the crystals.

*The section on IP*_*6*_
*regulation is fairly lengthy given the degree of speculation. The authors could consider condensing this*.

The molecular mechanism by which IP6 causes transient dimerization of Btk remains puzzling because we do not detect dimers even at concentrations as high as 30 μM. This section makes two points that we believe are important: (1) the conformation of the unliganded, monomeric Btk is likely to be different from that seen in the Saraste dimer and (2) changes in surface electrostatics can activate Btk in the absence of IP6. For these reasons we would prefer to retain this section.

*In the beginning of the Results and Discussion, it could be helpful to include a short introduction indicating which structures were determined as part of this work, to what resolution, and a reference to*
Table 2.

We thank the reviewers for this suggestion. We have added a section in the beginning of the Results and Discussion summarizing our crystallographic work (see ”Structural analysis of Btk”).

*It would be interesting to know whether the orientation of the IP*_*6*_
*in the peripheral site is well defined or not. IP*_*6*_
*binding to such a 'flat' (as the authors put it) site is often heterogeneous in pose, but the data in*
Figure 5
*suggest that this might not be the case here, as does the second paragraph in “The peripheral binding site is critical for IP*_*6*_*-mediated Btk activation”. Some comment on stereospecificity would be helpful for those who think about IP*_*6*_
*binding sites. Perhaps a density panel could be added to*
Figure 7
*or its supplement?*

We thank the reviewers for pointing out this important issue, which was not emphasized enough in the original manuscript. The disposition of the peripheral IP_6_ molecule can be determined unambiguously from the electron density because the phosphate group at position 2 juts out from the plane of the myo-inositol ring. This agrees well with our biochemical data that the phosphate at position 2 is likely to be critical for IP_6_ activation of Btk. We have added a figure showing electron density for IP_6_ at the peripheral site in Figure 7, and further explained the specificity of IP_6_ for this site in the Results section.

*In reflecting on the organization of the paper, one suggestion would be to place*
Figure 2
*(“Structure of a PH-TH-kinase construct of Btk”) before*
Figure 1
*(“Crystal structure of the Btk SH3-SH2-kinase module”). The story might flow/unfold better if the discovery of the potential autoinhibitory role of the PH-TH region leads into discovery of the Src-like disposition of domains in the Src-like module, then wrapped up nicely by the work in*
Figure 3
*on whole Btk*.

We appreciate this point. We have found it to be challenging to weave all the threads of the discussion into one coherent narrative. At the end we find it easier to stay with the current organization.

In addition to these changes made to the manuscript, in response to the reviewers’ comments, we have added one new figure (Figure 9—figure supplement 1). We used a simple Michaelis-Menten scheme, shown in this figure, to model the autophosphorylation kinetics of Btk. We make two assumptions in this scheme: first, that IP_6_ binding increases the on-rate of dimer formation of Btk by a factor of 10, and that once Btk is phosphorylated, the catalytic rate constant for transautophosphorylation increases by a factor of 10. Using physically reasonable values for the on and off rates and the catalytic rates we show that a small increase in on-rate for Btk dimer formation leads to a dramatic increase in the rate of autophosphorylation, even though the concentration of Btk is two orders of magnitude below the value of the dissociation constant. We think that this simple calculation helps explain an observation that might otherwise be counter-intuitive to some readers, which is that IP_6_ can activate Btk at Btk concentrations that are well below the dissociation constant for dimerization. Text referring to this new figure is in the last paragraph of “Fusing the Btk PH-TH module to the catalytic module of c-Abl promotes autophosphorylation of c-Abl in the prescence of IP_6_”.